# Towards a reliable assessment of climate change impact on droughts in Southern Italy: Evaluation of EURO-CORDEX historical simulations by high-quality observational datasets

David J. Peres[1], Alfonso Senatore[2], Paola Nanni[1], Antonino Cancelliere[1], Giuseppe Mendicino[2], Brunella Bonaccorso[3]

[1]Department of Civil Engineering and Architecture, University of Catania, Catania, 95123, Italy
[2]Department of Environmental Engineering, University of Calabria, Arcavacata di Rende (CS), 87036, Italy
[3]Department of Engineering, University of Messina, St Agata, Messina, 98166, Italy

*Correspondence to*: Alfonso Senatore (alfonso.senatore@unical.it)

**Abstract.** Many recent studies indicate climate change as a phenomenon that significantly alters the water cycle in different regions worldwide, also implying new challenges in water resources management and drought risk assessment. To this end, it is of key importance to ascertain the quality of Regional Climate Models (RCMs), which are commonly used for assessing at proper spatial resolutions future impacts of climate change on hydrological events. In this study, we propose a statistical methodological framework to assess the quality of the EURO-CORDEX RCMs concerning their ability to simulate historic climate (temperature and precipitation) and drought characteristics (duration, accumulated deficit, intensity and return period) determined by the theory of runs, at seasonal and annual time scales, by comparison with high-density and high-quality ground-based observational datasets. In particular, the proposed methodology is applied to Sicily and Calabria regions (Southern Italy), where long historical precipitation and temperature series were recorded by the ground-based monitoring networks operated by the formerly Regional Hydrographic Offices, whose density is considerably greater than observational gridded datasets available at the European level, such as E-OBS or CRU-TS. Results show that among the more skilful models, able to reproduce, overall, precipitation and temperature variability, as well as drought characteristics, many are based on the CLM-Community RCM, particularly in combination with the HadGEM2 GCM. Nevertheless, the ranking of the models may slightly change depending on the specific variable analysed, as well as the temporal and spatial scale of interest. From this point of view, the proposed methodology highlights the skills and weaknesses of the different configurations and can serve as an aid for selecting the most suitable climate model for assessing climate change impacts on drought processes and the underlying variables.

## 1 Introduction

A growing number of scientific studies claims that climate change due to global warming will significantly alter the water cycle, with an increase of the intensity and frequency of extreme hydro-climatic events in several areas around the globe

(Arnell et al., 2001; Huntington, 2006; IPCC, 2014; IPCC, 2018). These include the Mediterranean region, which is recognized as one of the major hot spots of climate change due to future projections of temperature increase and annual precipitation decrease (Giorgi, 2006; Kjellström et al., 2013).

Global Circulation and Regional Climate Models (GCMs and RCMs) can play a crucial role in understanding the potential spatiotemporal evolution of climate change in the future, thus improving current monitoring and planning tools (e.g., Mendicino and Versace, 2007; Hart and Halden, 2019) and supporting decision-makers to choose and implement the best solutions to minimize the impact of climate change on human systems and the environment at the regional scale. While GCMs' simulations describe climate evolution at large scale, by using coarse resolution information, RCMs simulations, derived

through climate-downscaling techniques, aim at representing regional and local scale weather conditions with grid resolutions lower than 50 km down to about 10 km (Kotlarski et al., 2014; Peres et al., 2019).

Several studies, focused on the use of climate models to simulate future climate scenarios for hydrological analyses, have shown that changes in temperature and precipitation vary in space depending on the future climate scenario, type, and resolution of the models, as well as on spatial heterogeneity of climatic features. This is particularly evident in the

Mediterranean region where, for instance, precipitation is partially controlled by orography, shows strong seasonality and large interannual fluctuations, and is characterized by the occurrence of particularly intense extreme events, such as prolonged droughts and high-intensity storms leading to floods (Bonaccorso et al., 2013; Bonaccorso et al., 2015a and 2015b; Llasat et al., 2016; Senatore et al., 2020).

Recently, there is a growing interest in the implementation of RCMs derived by dynamical downscaling of GCM outputs for

climate change impact studies at small spatial scales. These are high-resolution models able to provide a more realistic representation of important surface heterogeneities (such as topography, coastlines, and land surface characteristics) and mesoscale atmospheric processes.

The Coordinated Regional Climate Downscaling Experiment (CORDEX) initiative is the first international program providing a common framework to simulate both historical and future climate at the regional level, under different Representative

Concentration Pathways (RCPs) (van Vuuren et al., 2011), and over different domains which cover all the land areas. More specifically, it provides climate data simulated by an ensemble of RCMs developed by several research centres all over the world that are forced by Global Circulation Models (GCMs) from the Coupled Model Intercomparison Project phase 5 (CMIP5; Taylor et al., 2012). In the present study, we refer to the CORDEX domain centred on the Euro-Mediterranean area, known as EURO-CORDEX (Jacob et al., 2014; www.euro-cordex.net). In particular, EURO-CORDEX provides simulations

for a historic reference period (baseline) and future projections up to 2100, with a 12.5 km grid resolution, available for four RCPs defined at the international level within CMIP5.

The reliability of individual RCMs in representing climate effects on the hydrological cycle depends on the quality of historical simulations and must be evaluated before using their output for impact assessment. Assessing RCMs performance is essential to either select single models for further applications (e.g., Senatore et al., 2011; Peres et al., 2017; Smiatek and Kunstmann,

2019) or properly weight individual RCMs in multi-model ensembles to predict future impacts of climate change on

hydrological processes (e.g., Christensen et al., 2010; Coppola et al., 2010). Table 1 provides a broad, although not thorough, list of intercomparison studies within the CORDEX framework available in the literature. Overall, these studies show that CORDEX RCMs can reproduce the most important climatic features at regional scales, but important biases remain, especially regarding precipitation or climate extremes. As reported by Kotlarski et al. (2014) and references therein, model biases may depend on the analysed region, choices in model configuration, internal variability, and uncertainties of the observational reference data themselves (Gampe et al., 2019). Concerning the latter, a common approach in evaluation exercises consists in comparing models' simulations to observational gridded datasets, from remote sensing or model-derived reanalyses products available at global or continental spatial scales.

In general, statistical measures, such as bias, root mean square error, correlation, and trend analysis, are used to quantify model performance. Regardless of the specific methods used to assess the differences between simulated and observed data, one of the main limitations in this approach is that the considered spatial resolution is too coarse for reliable climate change impact studies at relevant hydrological scales, especially in areas of complex topography. From this point of view, large-scale observational gridded datasets are of poor applicability, since they are built upon low-density hydro-meteorological networks. In principle, more accurate evaluations can be achieved when they rely on gridded reference data sets that are obtained by spatial interpolation of point measurements onto a regular grid. To this end, two main prerequisites are that data coverage well reflects the topography and variables with limited spatiotemporal climatic variability are investigated (Wagner et al., 2007). For example, Mascaro et al. (2018) compared the skill of several EURO-CORDEX RCMs at ~ 50 and 12 km grid spatial resolution in reproducing annual and seasonal precipitation regimens and trends in Sardinia (Italy), against a dense network of rain gauges with long term records. Their analysis revealed that, although the simulated spatial patterns of annual and seasonal means are well correlated with the observations, positive and negative biases up to ±60% in the simulation of annual mean and interannual variability are detected. Furthermore, the majority of RCMs underestimate winter and overestimate summer precipitation.

In this study, we present an enhanced analysis over a different Mediterranean area with complex topography, namely Sicily and Calabria regions (Southern Italy). In particular, after investigating the ability of the EURO-CORDEX models to simulate the annual and seasonal temperature and precipitation regime, we analysed the skill in reproducing drought event characteristics identified through the run method (Yevjevich, 1967). Within the drought analysis, we also investigated the return period of drought events of fixed duration at both the annual and seasonal scales. In this case, given the limited number of droughts in a thirty-year long time series, an analytical framework was applied that allow computing return period based on reasonable assumptions on the probabilistic structure of annual and seasonal precipitation (Bonaccorso et al. 2003; Cancelliere and Salas, 2004). Furthermore, we analysed model skills at a sub-regional level. To this aim, we proposed the use of Principal Component Analysis (PCA) for delimitation of climatically homogeneous areas. The ability of climate models to reproduce observed precipitation, temperature and drought features was analysed both per single characteristic as well as per multiple characteristics (e.g. precipitation and temperature together), by introducing a specific ranking criterion.

Nineteen coupled GCM and RCM simulations within the EURO-CORDEX framework were evaluated against a high-density and high-quality monitoring station-based reference dataset. Monthly temperature and precipitation records were retrieved by two monitoring networks, operated by the former Regional Hydrographic Services, whose density is significantly higher than observational datasets available at the European scale, such as E-OBS (Haylock et al., 2008) or CRU-TS (Harris et al., 2014), allowing for a more accurate evaluation of the models.

## 2 Study area and datasets

Our analyses were focused on Calabria and Sicily regions in Southern Italy, which respectively have an extension of 15,080 km² and 25,460 km², for a total area of 40,540 km² (Fig. 1). Climate is of Mediterranean type with hot and dry summers and moderately cold winters with peak monthly precipitation occurring mostly in late autumn and winter. About 75% of the total precipitation in the study area occurs from October to March, because of cyclonic storms. These climate features make the area particularly prone to droughts, with the most recent severe episode occurred in 2017 (Rossi and Benedini, 2020). Climate features are also highly variable in space due to a rather complex orography. In particular, the mountain chains close to the coast enhance intense orographic precipitation and lead to relatively cold temperatures at the highest altitudes.

### 2.1 Observed data

Within the EURO-CORDEX control period (1951-2005), the comparison with observations was performed in the period from 1971 to 2000. These three decades had the greatest availability of historical series of precipitation and temperature recorded by both the regional monitoring networks of Calabria and Sicily, managed by the Multirisk Operational Centre of Calabria region (ArpaCal) and the Water Observatory of Sicily region (WOS), respectively. Specifically, 84 thermometers (43 in Sicily and 41 in Calabria Calabria and near the regional borders) and 335 rain gauges (173 in Sicily and 162 in Calabria and near the regional borders) were used (Fig. 1). Details on the monitoring network are given in the Supplementary Material to this paper. The corresponding data were retrieved by the WOS (www.osservatorioacque.it) and the ArpaCal (www.cfdcalabria.it) websites. Observations were enough widespread to represent the quite heterogeneous features of the study area. The temperature stations were located between 2 and 1295 m a.s.l., with annual average values ranging from 9.2 °C to 20.6 °C (mean value = 16.2±2.4 °C), while the rain gauge elevations varied from 1 to 1369 m a.s.l., with annual accumulated values ranging from 373 mm to 1736 mm (mean value = 812±287 mm).

### 2.2 Climate models

Monthly precipitation and monthly mean air temperature data from the EURO-CORDEX CMIP5 simulations (Jacob et al. 2014; https://www.euro-cordex.net/) were retrieved from the nodes of the Earth System Grid Federation (ESGF, e.g. https://esgf.llnl.gov).

We analysed the data at the finest resolution, 0.11° (~ 12.5 km), EUR-11 and considered the period 1971-2000 as a baseline. In particular, the combination of six GCMs (Tab. 2) and eight RCMs (Tab. 3) leading to 17 datasets, reported in Tab. 4, were collected for the study. Moreover, for two GCM-RCM combinations, two versions were available from the ESGF portal. Therefore, an overall ensemble of 19 combined models (CMs) was analysed. The ensemble mean of the 19 CMs was also evaluated. Even if the CMs have the same spatial resolution, each one is distributed on a specific grid (with slightly different origin and orientation of the axis). Therefore, the various data sets were resampled on the grid of the ECE-HIRH CM, which is shown in Fig. 1.

We choose EUR-11 rather than EUR-44 simulations as several studies (Torma et al., 2015; Prein et al., 2016) have found that generally higher resolution CORDEX RCMs have better skills in simulating seasonal precipitation in regions with complex terrain.

## 3 Methodology

### 3.1 Data processing and PCA

To allow the comparison between the spatially distributed RCMs data and site-specific observations, the latter were spatially interpolated using the CORDEX 0.11° grid as reference (Fig. 1). In this way, month by month, each cell of the CORDEX grid could be associated with a single temperature or precipitation value derived from the observations network. Specifically, concerning temperature, an Inverse Distance Weighting (IDW) interpolation was applied to the residuals of the values obtained using a regression model with the altitude. For precipitation, whose measurement network is much denser, a simple IDW interpolation was performed. As shown in Fig. 1, the CORDEX grid cells which are not covered by any rain gauge are relatively few (less than 30%) and, except one case, the distance of the closest rain gauge to every grid cell is always less than 10 km.

The precipitation patterns obtained by the interpolation procedure were analyzed adopting a methodology based on the Principal Component Analysis (PCA) to distinguish zones with rather independent climatic variability within the area under investigation. PCA is a well-known statistical tool used to transform an original set of intercorrelated variables into a reduced number of new linearly uncorrelated ones explaining most of the total variance (Rencher, 1998). The latter, derived as linear combinations of the original variables, are the principal components (PCs), while the coefficients of the linear combinations are the loadings, which in turn represent the weight of the original variables in the PCs. From a procedural standpoint, PCA consists of solving an eigenvalue-eigenvector problem applied to the covariance matrix. The eigenvectors, properly normalized, are the loadings of the principal components, while the eigenvalues provide a measure of the total variance explained by each loading (Bordi and Sutera, 2001 and references therein). Under this decomposition, the loadings represent the correlation between the associated PCs and observed time series. Mapping the loading patterns of each PC among those selected, based on the percentage of the total explained variance of the process, largely allow to identify independent climatic areas within the study region. Moreover, it may be useful to apply a rotation operation to the eigenvectors, so that the corresponding loadings are more spatially localized. In other words, the rotation leads to loadings with a high correlation with

a smaller set of spatial variables and a low correlation with the remaining variables. Here, only orthogonal rotations were considered, computed by the varimax algorithm in Matlab® R2016. Clearly, each rotated pattern will not explain the same variance of the unrotated one, although the total variance explained remains unchanged.

In the present study, the first nine rotated PCs both at the annual and seasonal (DJF, MAM, JJA, SON) scales were investigated. Regardless of the considered period, the selected PCs always explain more than 78% of the total variance, with a maximum of

85% in the winter season (DJF). The loading patterns of these rotated PCs were mapped for each considered period to identify climatically homogeneous regions. Homogeneous sub-regions were detected at the annual scale and in autumn and winter seasons, whereas a confusing picture arose in spring and summer seasons. Furthermore, since about 75% of the total annual rainfall of the case-study area occurs between autumn and winter (as a result of cyclonic storms), the climatically homogeneous sub-regions identified at the annual scale approximately overlap with those identified at seasons SON and DJF. Isolated grid

cells showing a different PC correspondence with respect to the surrounding cells, were manually corrected to simplify the delimitation of the homogeneous sub-regions. This approach led in dividing the whole area into six climatically homogenous zones, three for Sicily and three for Calabria (Fig. 1), for which separate performance assessments were carried out. Concerning Sicily region, the three identified sub-regions roughly coincide with the ones detected by Bonaccorso et al. (2003), who investigated the spatial variability of droughts in Sicily region based on SPI series computed on monthly precipitation observed

in traditional rain gauges and on NCEP/NCAR reanalysis data from 1926 to 1996. In particular, three distinct areas, namely North-Eastern (identified in the PCA as zone 5, Fig. 1b), South-Central Eastern (zone 4), and Central-Western (zone 1), were identified. In Calabria, three main zones were also determined, namely North-Western (zone 2), North-Eastern (zone 3) and South-Eastern (zone 6), broadly corresponding to climatic homogenous areas found in previous studies (e.g., Versace et al., 1989). Interestingly, the South-Western tip of Calabria is identified as a part of a broader area (zone 5) extending over the

North-Eastern Sicily.

### 3.2 Performance metrics and models' ranking

The CMs were evaluated based on their performances in capturing specific properties, namely: the interannual and seasonal variability of precipitation, temperature and drought characteristics. Such properties were expressed based on some relevant statistics.

Let $X(j)$ and $X_\tau(j)$ be the variable under investigation (precipitation or mean temperature) at grid cell $j$ at the annual and seasonal scale, respectively. For precipitation and mean air temperature, the following statistics were derived for each CM and cell in the area of interest:

- Seasonal mean $\mu_m\big(X_\tau(j)\big) = \frac{\sum_{i=1}^{N} x_{\tau,i,m}(j)}{N}$

  where $x_{\tau,i,m}(j)$ is the value of the variable at season $\tau$ ($\tau = 1, 2, 3, 4$) and year $i$ ($i = 1, 2, \dots N$) produced by the $m$-

th CM ($m = 1, 2, \dots M$) at cell grid $j$. Seasons are December – February (DJF), March – May (MAM), June – August (JJA), and September – November (SON);

- Seasonal standard deviation $\sigma_m\left(X_\tau(j)\right) = \sqrt{\dfrac{\sum_{i=1}^{N}\left(x_{\tau,i,m}(j)-\mu_m(X_\tau(j))\right)^2}{N-1}}$;

- Annual mean $\mu_m\left(X(j)\right) = \dfrac{\sum_{i=1}^{N} x_{i,m}(j)}{N}$;

  where $x_{i,m}$ is the value of the variable at year $i$ ($i$=1, 2, … $N$) produced by $m$-th CM;

- Annual standard deviation $\sigma_m\left(X(j)\right) = \sqrt{\dfrac{\sum_{i=1}^{N}\left(x_{i,m}(j)-\mu_m(X(j))\right)^2}{N-1}}$.

Drought events were identified on both annual and seasonal (DJF, MAM, JJA, SON) precipitation values simulated for the period 1971-2000, according to the theory of runs (Yevjevich, 1967). In particular, drought events were selected as the periods during which consecutive annual or seasonal values of precipitation did not exceed a given threshold, here assumed equal to the long term means of annual and seasonal data (i.e. one threshold for each season). Once drought events were identified, the
corresponding drought characteristics in each cell were determined. In particular, the following statistics for drought characteristics are considered hereafter to assess the models' performance:

- Maximum drought duration $L_{max}$: maximum length of periods with consecutive annual precipitation values below the threshold;

- Maximum drought accumulated deficit $D_{max}$: maximum of the sums of the differences between the threshold and the
precipitation values along with the drought duration;

- Maximum drought intensity $I_{max}$: maximum of the ratio between drought accumulated deficit and duration;

- Return period of drought events of fixed duration (at both annual and seasonal scales).

Concerning the return period of drought events, let $E$ be a critical drought (e.g., a drought with duration $L$ equal to a fixed value). Assuming independence between consecutive drought events, the return period of drought event $E$ can be expressed as
(Gonzales and Valdes, 2003; Cancelliere and Salas, 2004; Cancelliere and Salas, 2010; Bonaccorso et al., 2012):

$$T_E = \frac{E[L]+E[L_n]}{P[E]} \tag{1}$$

where $E[L]$ is the expected value of drought duration $L$ and $E[L_n]$ is the expected value of the non-drought duration $L_n$ and
$P[E]$ is the probability of occurrence of a critical drought $E$, which can be determined once that the probability distribution function of the event $E$ is known.

Regarding the probability distribution of drought duration, let us consider a stochastic hydrological variable denoted as $X_{\nu,\tau}$, where $\nu$ represents the year and $\tau$ represents the season. According to the theory of runs, drought duration $L$ is the number of consecutive time intervals (seasons) where $X_{\nu,\tau} \leq x_{o,\tau}$ is preceded and followed by at least one season where $X_{\nu,\tau} >$
$x_{o,\tau}$, where $x_{o,\tau}$ is a threshold level representing water demand. The original variable can be replaced by a Bernoulli variable $Y_{\nu,\tau}$ such that:

$$\begin{cases} Y_{v,\tau} = 0 \; if \; X_{v,t} \le x_{0,\tau} \; (deficit) \\ Y_{v,\tau} = 1 \; if \; X_{v,t} > x_{0,\tau} \; (surplus) \end{cases} \tag{2}$$

Assuming that $Y_{v,\tau}$ is a lag-1 Markov stationary process, it can be shown (Sen, 1976; Cancelliere and Salas 2004; Cancelliere and Salas, 2010) that the probability distribution of drought duration $L$ is geometric with parameter $p_{01}$:

$$f_L(\ell) = P[L == P[Lp_{01})^{\ell-1} p_{01} \tag{3}$$

The parameter $p_{01}$ represents the transition probability from a deficit to a surplus, namely $p_{01} = [Y_{v,\tau} = 1|Y_{v,\tau-1} = 0]$. Estimation of transition probabilities can be carried out following a non-parametric approach based on maximum likelihood, which leads to (Bonaccorso et al., 2012):

$$p_{01} = 1 - p_{00} = 1 - \frac{n_{00}}{n_{00}+n_{01}} \tag{4}$$

where $n_{00}$ is the number of observations $y_{v,\tau} = 0$, for which $y_{v,\tau-1} = 0$, and $n_{01}$ is the number of observations $y_{v,\tau} = 1$, for which $y_{v,\tau-1} = 0$.

For independent stationary series, the probability distribution of drought duration $L$ is geometric with parameter $p_1 = P[Y_\tau = 1]$. The latter can be simply estimated by applying a frequency analysis on $Y_\tau$.

Following previous studies (Bonaccorso et al., 2003; Cancelliere and Salas, 2004), the annual series were assumed independent stationary, whereas the seasonal series as lag-1 stationary Markov.

Models' skills in reproducing the interannual and seasonal variability of precipitation and mean air temperature variables were first assessed through:

- boxplots of the errors and percentage errors of the mean values in all the grid cells of the investigated areas, which
allow analysing the spatial variability of the models' bias;
- Taylor diagrams (Taylor, 2001), which show three metrics at the same time, i.e.: coefficient of correlation, standard deviation, and centred root mean square error of the anomalies (i.e., the variables of interest minus the corresponding means). It is noteworthy that standard Taylor diagrams do not provide any information about first-order statistics (i.e., bias).

Later, to provide synthetic information about each CM starting from the various statistics computed for each property, a method based on Mascaro et al. (2018) was used. Specifically, for each property (i.e. seasonal and interannual variability of

precipitation and mean temperature and drought characteristics), a single dimensionless error metric that combines multiple statistics characterizing that property was estimated. The error metrics follows the equation:

$$\varepsilon_m = \sqrt{\sum_{k=1}^{S} \left( \frac{\sum E_{k,m}(j)}{\sum_{m=1}^{M} \sum_{j=1}^{P} E_{k,m}(j)} \right)^2}$$

(5)

where $E_{k,m}(j)$ represents an error metric between observed and simulated data of the statistics $k$ ($k = 1,\ldots, S$) at grid cell $j$ ($j=1, \ldots P$, where $P$ is the total number of grid cells), whose sum over the whole area was divided by the sum of the error metrics of all models, therefore resulting in a dimensionless indicator for each statistic $k$ of any property. Table 5 summarizes the statistics chosen for each property and describes how the corresponding errors were calculated.

Based on the values of the error metrics in Eq. (1), a ranking of the models, describing the skills in reproducing each property, was obtained. It should be specified that while, for the sake of brevity, the boxplots and the Taylor diagrams illustrated in the next section refer to the whole study area, the ranking of the models for the mean air temperature, precipitation and drought characteristics also refers to the six climatically homogenous zones identified through PCA. This analysis, indeed, can help to highlight whether some models are more suitable than others to simulate certain variables in a given zone.

## 4 Results

In this section, results are presented and discussed separately for temperature, precipitation and drought characteristics. Results are differentiated for the following temporal and spatial aggregation scales: annual data, seasonal data, the whole case study region and the six climatically homogenous areas identified via PCA.

### 4.1 Mean air temperature

#### 4.1.1 Interannual variability

The observed and modelled means of the annual mean air temperature values in each of the grid cells within the study area were calculated and compared. More specifically, for each cell $j$, the error corresponding to the $m$-th CM was computed as:

$$E_{m,j} = \mu_m(T(j)) - \mu_0(T(j))$$

(6)

where $T(j)$ is the mean annual temperature at cell $j$, whereas $\mu_m(\cdot)$ and $\mu_0(\cdot)$ are the modelled and observed means respectively. For each model, the distribution of the errors computed for all the grid cells of the study area based on Eq. (2), is represented in the form of box-plots in Fig. 2a. In particular, the central line represents the median value and the box is delimited by the first and the third quartile. The width of the box corresponds to the inter-quartile range (IQR), a well-known measure of dispersion. Values outside the whiskers, distant from the box at least 1.5 IQR, can be assumed as outliers.

The overall tendency of the models is to underestimate temperatures, as the medians are negative. Errors are predominantly comprised between the values -5 and -1 °C, thus implying that the models underestimate up to 5 °C. The CMs that produce

the most extreme negative errors are the ECE-RACM, ECE-RACMr12 and CM5-ALAD, with the latter showing the broader IQR (e.g. the highest spatial variability of the errors) and the greatest median error. All the CMs with RCA4 show the smallest IQR. The models with the smallest median error are MPI-REMO and MPI-REMOr2.

To extend the CM skill comparison to other statistics, the Taylor diagram for the annual mean air temperature values was developed (Fig. 2b). For the sake of simplicity, standard deviations of the CMs are indicated as σ hereinafter. The diagram allows visualizing if there are clusters of performances related to specific GCMs or RCMs among those considered. In the diagram, GCMs are indicated with different markers, while RCMs with different colors. The value corresponding to the observations is the dot on the *x*-axis, whose standard deviation is marked through a continuous circular arc. In addition to every single model, the ensemble mean model result is reported in the diagram.

Fig. 2b shows that the simulated means are well correlated with the observations, with values larger than 0.8 for all the considered models. Furthermore, the diagram seems to reveal that, on equal GCMs, RCMs play a significant role in determining the performance of the combinations. In general, for most of the models, the best performances are obtained when the RCM RCA4 is used. The only exception is CM5, performing better in combination with CCLM. The worst models are CM5-ALAD and IPS-WRF.

Finally, the ranking analysis described in Section 3.2 yields the results in Fig. 3. The lower the rank, the lower is the error metrics in Eq. (1) and the better is the model. For better readability, ranking values are indicated through a chromatic scale, ranging from dark green (first ranked model) to dark red (last ranked model).

The best performing models, in terms of ranking order for the whole study area, are MPI-CCLM, MPI-REMO, and Had-CCLM. ECE-RCA4 and CM5-CCLM are also good models as highlighted by the Taylor diagrams. Figure 4 also shows rankings for each of the six homogeneous areas. As it can be observed, based on the range of colours in each row, MPI-CCLM and MPI-REMO provide the best performance for almost all the zones.

Indeed, some differences exist for Zones 3 and 6 (North and South-Eastern Calabria), whose best CM is IPS-RCA4. Overall, results show that the worst model is CM5-ALAD for entity and dispersion of errors, lower correlations, higher RMSE, greater deviation from the standard deviation of the observed values, both for the whole study area and individual zones. ECE-RACM, ECE-RACMr12, and ECE-RCA4 also show bad performance (the latter mainly because of its relatively strong bias).

**4.1.2 Seasonal variability**

For the sake of brevity, the box-plots related to the seasonal variability of mean air temperature are not shown since they provide similar results to the case of annual variability.

Figure 4 shows the Taylor diagrams obtained from the analysis of the individual seasons. CM5-ALAD and IPS-WRF (and, to a slightly lesser extent, CM5-ALAR) appear as the worst models regardless of the season, although in summer (JJA) the worst-performing models are MPI-REMO and MPI-REMOr2. Summer is also the season with the (slightly) lowest values of correlation coefficients.

Regarding the best models, in general, all the combinations with RCA4 and the CM5-CCLM work better, as for the interannual variability analysis. However, in summer better performances are obtained with ECE-RACM and ECE-RACMr12.

Figure 5 represents the rankings of the models for the individual seasons and all the study areas, namely the whole case study and the six zones. There is a certain correspondence on the least performing models between Figs. 4 and 5. Nonetheless, differently from the results in Fig. 3, models' performances may change significantly from season to season and, in the same season, from zone to zone. The best models for most of the zones are ECE-HIRH in winter (DJF), ECE-CCLM in spring (MAM), IPS-RCA4 in summer (JJA) and MPI-REMOr2 in autumn (SON). It's worth highlighting that the latter provides the best performances also for Zones 2 and 4 in spring and Zones 5 and 6 in summer. Conversely, ECE-HIRH, which is the best model in winter, works poorly in summer and autumn. The Zones 1 (Western Sicily) and 2 (Western Calabria) show a uniform behaviour in all seasons, with the only exception of spring, while Zones 5 (North-Eastern Sicily) and 6 (South-Eastern Calabria) show a uniform behaviour in all seasons but autumn. Besides, in summer and autumn, the best performing models for Zones 1, 2 and 4 (South-Eastern Sicily) are the same as for the whole study area. Zone 3 (North-Eastern Calabria) behaves like Zone 4 in winter and like Zones 1, 5 and 6 in spring.

## 4.2 Precipitation

### 4.2.1 Interannual variability

Figure 6a shows box-plots for the percentage errors in mean annual precipitation, namely:

$$E_{m,j} = \frac{\mu_m(P(j)) - \mu_0(P(j))}{\mu_0(P(j))} \cdot 100 \tag{7}$$

where $P(j)$ is the total annual precipitation at the grid cell $j$.

In comparison to temperature, the errors are much larger, as well as the differences between the various models. There is a general tendency for the models to underestimate the total annual precipitation, except for some models like IPS-WRF, which also shows the largest IQR. The median value of the relative errors for some models is less than 20%; however, many models have a large dispersion with error values over 100%. The CM with the highest positive error is IPS-WRF, while the ones with the highest negative errors are the IPS-RCA4 and Nor-HIRH models. The GCM-RCM combinations with the smallest IQR of errors are those using CCLM RCMs. The model with the smallest bias is Had-RACM.

The Taylor diagram in Fig. 6b confirms that the best combinations are those with CCLM RCMs. In particular, the best one seems ECE-CCLM. However, when used in combination with CM5, the corresponding model provides poor performance. The worst performing models are ECE-HIRH and Nor-HIRH. The diagram confirms that precipitation is modelled with less accuracy than temperature, as correlations are lower (<0.8).

The application of the ranking criteria (see Fig. 7) suggests Had-RACM and ECE-CCLM as the best combinations for the entire area and most of the zones. Also, CM5-ALAD works well for the whole area and almost all the zones, except for Zone 4, where it ranks the 11th. IPS-WRF, IPS-RCA4, Nor-HIR, and CM5-RCA4 are the worst models.

#### 4.2.2 Seasonal variability

The seasonal variability analysis carried out on precipitation shows (Fig. 8) a lower error dispersion in the wet seasons (i.e., autumn and winter) with respect to summer. In summer, several models show broader IQR, such as all the CM5 models and IPS-WRF, with the latter showing the largest median error. On the one hand, these outcomes depend on the poor performance of some models in reproducing the seasonal cycle, and on the other hand, are due to the fact that in the dry season where rainfall is normally low, large errors may result even though the departure from the observed mean is relatively small. These results are consistent with those obtained by Giorgi and Lionello (2008) in a subdomain of the Mediterranean region and by Mascaro et al. (2018) for the Sardinia region.

The Taylor diagrams in Fig. 9 highlight that NOR-HIRH and ECE-HIRH are the worst models for all the seasons but summer, where the IPS-WRF is the worst-performing.

These indications are confirmed by the ranking results in Fig. 10. Concerning the best models, the following CMs perform the best in their respective seasons: ECE-RACMr12 in winter (DJF), ECE-CCLM in spring (MAM), MPI-REMOr2 in summer (JJA), MPI-CCLM and Had-RACM in autumn (SON). It is worth highlighting that ECE-RACMr12 provides the best rank also for Zone 2 in autumn; ECE-CCLM is the best performing also for Zone 6 in summer; MPI-CCLM provides the best performances also for Zone 1 in winter and Zone 4 in spring and Had-RACM is the best model for Zone 2 in spring. For summer precipitation, MPI-REMOr2 is the best performing CM also for Zones 1, 2, 3 and 4. As for the ranking of seasonal mean temperature, once again there is no uniform behaviour of the models between the different seasons and zones.

### 4.3 Drought characteristics

#### 4.3.1 Annual scale

The models' performance in reproducing historical drought characteristics both at the annual and the seasonal scale was also tested. In particular, the following drought characteristics derived from the theory of runs were analysed: maximum duration ($L_{max}$), maximum accumulated deficit ($D_{max}$), and maximum intensity ($I_{max}$) and return period of drought duration.

With reference to the drought characteristics identified on annual precipitation, Figures 11a, b and c represent the boxplots of the errors related to maximum drought duration, accumulated deficit, and intensity, respectively. In particular, for drought duration, the errors were computed through Eq. (2) by simply replacing $T$ with $L_{max}$, whereas for maximum drought accumulated deficit and intensity, the percentage errors were calculated through Eq. (3), by replacing $P$ first with $D_{max}$ and then with $I_{max}$.

There is a slight tendency of some models to underestimate drought duration (Fig. 11a). Overall, the errors span from -3 and +2 years. The broadest IQR is associated with MPI-REMO, while some models, such as CM5-CCLM, CM5-ALAR, ECE-RACM and, Nor-HIRH seem equally reliable.

The boxplots obtained for $D_{max}$ (Fig. 11b), show that the models may yield considerable errors, which can potentially be larger than those for annual precipitation, as the accumulated deficit, given by the sum of precipitation deficits on a time interval

lasting several years, can be affected by multiple errors. For some models, the IQRs are not larger than 50%. The most reliable model is Had-CCLM, but comparable performances are given by models CM5-CCLM, CM5-ALAR and ECE-CCLM, while the least dispersed is MPI-CCLM (for this model, however, the median error is larger than others). The least reliable is IPS-WRF, followed by CM5-RCA4 and MPI-REMOr2. In general, as it can be seen from the box-plots, this feature is underestimated. Concerning $I_{max}$, the results indicate Had-RACM as the best model and CM5-RCA4 as the worst, followed by

IPS-WRF (Fig. 11c). Errors for this feature are less scattered than for accumulated deficit, and there is a general tendency for $I_{max}$ to be underestimated by models.

Figure 12 shows box-plots of the errors in the return period of drought events of duration $L$ equal to 1, 3, 5 and 7 years, respectively. In particular, the error was calculated as:

$$E_{m,j} = \mu_m\left(T_y(j)\right) - \mu_0\left(T_y(j)\right) \tag{8}$$

where $T_y(j)$ is the return period of a drought event of fixed duration at the grid cell $j$.

As expected, on equal model, the error increases as the considered drought duration increases. However, regardless of the drought duration, there is no general tendency of the models towards overestimation or underestimation of the return periods. ECE-CCLM and Had-RACM are the models with the smallest IQR, with ECE-CCLM showing the lowest median error. Overall, the performance of the models looks rather similar, with limited errors until L=3 years (± 0.5 years).

Finally, the models were also ranked according to their ability in reproducing both observed drought maximum intensities and return periods of drought events with duration L=3 years (Fig. 14a). Drought intensity was selected as it merges drought accumulated deficit and duration of each drought event. Concerning the return period, it is worth pointing out that the choice of the considered drought duration only affects the magnitude of the errors, while the performance of each model with respect to the others does not change (see Fig. 12). As shown in Fig. 14a, the best models for the whole study area are confirmed to

be ECE-CCLM, Had-RACM, ECE-RACM, and Had-CCLM. Interestingly, CM5-ALAR is the best model for Zone 3, but unsuitable for the remaining zones. The worst model for all the zones is CM5-RCA4, whereas poor performances are associated to ECE-RACMr12 for Zones 1 and 2, Had-RCA4 for Zone 3, MPI-REMOr2 for Zones 4 and 6 and IPS-WRF for Zone 5.

**4.3.2 Seasonal scale**

Figures 11d, 11e and 11f represent the boxplots of the errors related to maximum drought duration, accumulated deficit, and

400 intensity identified on seasonal precipitation data.

Concerning drought duration (Fig. 11d), several models (9 out of 19) show a median error equal to 0, while the other models tend to underestimate, with the only exception of IPS-WRF. Overall, the errors span from -4 and +3 seasons. The broadest IQR is associated with CMC5-ALAR and ECE-CCLM, while some models, such as IPS-RCA4, MPI-RCA4, MPI-REMOr2 and, Nor-HIRH seem equally reliable.

As for $D_{max}$ (Fig. 11e), some similarities can be observed concerning the annual time scale (Fig. 11b) in terms of magnitude of percentage errors, although in the seasonal case most of the models tend to overestimate. The most reliable models are

CM5-ALAD, ECE-CCLM and Had-RACM. As for the annual scale, the least reliable is IPS-WRF, followed by CM5-RCA4 and Nor-HIRH.

Concerning $I_{max}$, also in the seasonal case Had-RACM is confirmed as the best model, while MPI-REMOr2 and IPS-WRF are the worst (Fig. 11e). Once again, errors for this feature are less scattered than for accumulated deficit. Only four models underestimate $I_{max}$ while most of the models are close to a zero median percentage error.

Figure 13 shows box-plots of the errors in the return period of drought event of duration $L$ equal to 2, 4, 6 and 8 seasons, respectively. In particular, the error was calculated as in Eq. (8) by replacing $T_y$ with $T_s$, namely the return period of a drought event of fixed duration identified on seasonal data. As for the annual case, the performance of the models looks rather similar, with limited errors ($\pm$ 5 seasons) until L=4 seasons, with the exception of CM5-ALAD, CM5-ALAR, CM5-RCA4 and Had-RCA4.

Figure 14b illustrates the ranking of the models in reproducing the drought maximum intensities and return periods of drought events with duration L=4 seasons. With respect to the annual scale, there is a certain agreement in identifying the best performance models, that in this case are Had-RACM, Had-CCLM, and ECE-CCLM. In particular, Had-RACM performs well in every zone, while Had-CCLM is the best model for Zones 1, 2, 5 and 6. The least performing models are CM5-ALAD, CM5-ALAR, albeit it ranks second for Zone 5, CM5-RCA4 and Nor_HIRH.

## 5 Discussion

Table 6 illustrates the best performing models according to the ranking approach for each of the considered variables over the whole area and the six homogeneous zones, respectively. In particular, the three best performing models are reported for the mean temperature and precipitation interannual variability and drought intensity and return period of drought duration, while only the best CM for each season is indicated for seasonal variability.

It is worth underlining that the rankings are aimed to provide straightforward information about the relative accuracies of the models, e.g., for supporting the selection of a single or few models in a specific area, therefore, for the sake of simplicity, they provide reduced information based on cardinal numbering. However, the actual performance of each CM compared to the others can be highlighted by looking closer at the $\varepsilon_m$ values, which reflect and summarize the results provided by the box-plots and the Taylor diagrams.

Two kinds of comparisons are carried out in this section: 1) on the same variable, across different time scales; 2) on the same time scale, across different variables. Further discussion is provided about relative impacts of different GCMs and RCMs and, finally, an overall ranking is attempted aimed at providing a global evaluation of the CMs performance.

### 5.1 Analyses across different time scales (interannual and seasonal)

Concerning temperature, the intercomparison between the interannual and seasonal variability is rather straightforward. All the simulations are characterized by a more or less pronounced underestimation (Fig. 2a), together with a usually high

correlation with observations (Fig. 2b and 4), i.e. both the observed interannual and seasonal variability are well reproduced. This is somehow confirmed by the rankings, where the relative differences among the models' performances are not very marked.

Conversely, in the case of precipitation, the performances of the models change significantly with the time scale. The most interesting case with this variable is CM5-ALAD that, considering the total area, ranked $3^{rd}$ with the annual precipitation, but provided low performances in most of the seasons ($9^{th}$ in MAM, $11^{th}$ in DJF and $18^{th}$ in JJA). Though CM5-ALAD can reproduce relatively well the annual amount of rainfall, it is not as much able to simulate the seasonal variability, therefore the good performance at the annual time scale is due to the counterbalancing effects of the errors in different seasons. This feature of CM5-ALAD is amplified in several of the six zones, e.g., zone 2 (where it is ranked $4^{th}$ with the mean annual value, but $14^{th}$ in DJF and $18^{th}$ in MAM and JJA) or zone 6 ($1^{st}$ with the mean annual value, but $13^{th}$ on DJF and $18^{th}$ on JJA). On the other hand, MPI-CCLM in the total area ranked $8^{th}$ considering the annual precipitation but provided rather good results in single seasons (it is ranked $3^{rd}$ on MAM and $1^{st}$ on SON).

However, considering the total area and the annual precipitation, the values of the error metric $\varepsilon_m$ leading to the rankings are not very different among the first 9 models, being the $\varepsilon_m$ value of the model ranked $9^{th}$ (i.e., CM5-ALAR) only 37% higher than the best. The difference with respect to the best $\varepsilon_m$ value is lower than 50% in DJF for the first 7 models, in MAM for the first 5 models, in JJA for the first 6 models and in SON for the first 7 models. The models providing always (i.e., considering both the annual and the seasonal values) differences lower than 50% with respect to the best $\varepsilon_m$ value are Had-RACM, ECE-CCLM and Had-CCLM.

Figure 15 shows a comparison between the ranking of interannual variability of annual precipitation and the average position in the ranking of seasonal precipitation. It highlights possible deviations of the performances of the models at different time scales (the higher the deviation, the higher the distance from the bisector). When considering the seasonal scale, the reduced performance of CM5-ALAD is evident, such as the better ranking of MPI-CCLM. In general, the best models, both at the interannual and the seasonal scale, are Had-RACM and ECE-CCLM, followed by the two versions of ECE-RACM and two other CCLM models (namely, MPI-CCLM and Had-CCLM, the latter being penalized by the relatively lower ranking in winter).

Focusing on drought analysis, box-plots highlight a relevant variability in the frequency distribution of the error for all the considered drought characteristics. As for drought duration (Fig. 11a and d), the differences among the models appear more evident at the annual scale, while at the seasonal scale the models' behaviour looks rather similar. A general agreement can be observed between the box-plots of drought accumulated deficit at the annual and the seasonal scale (Fig. 11b and e), where the IPS-WRF is confirmed as the worst model. Concerning drought intensity (Fig. 11c and f), CM5-RC4 provides a very poor performance at the annual scale, but a light improvement can be observed at the seasonal scale.

As for the return periods (Fig. 12 and 13), the seasonal scale emphasizes the poor quality of CM5-ALAD, which is also confirmed at the annual scale, together with CM5-ALAR, ECE-RACMr12 and MPI-CCLM.

Finally, the rankings combining the performance of the models to simulate maximum drought intensity and return period of drought event of fixed duration (Fig. 14a and b) agree in considering Had-RACM and ECE-CCLM as the best models both at the annual and seasonal scale.

## 5.2 Analyses across different variables

In terms of interannual variability, it's worth observing that, while MPI models appear the most suitable for mean temperature regardless of the area of investigation, especially regarding those in combination with REMO and CCLM RCMs, this is not the case for precipitation, although both the boxplot and the Taylor diagram indicate some potential of the MPI-CCLM for precipitation (Fig. 6). The boxplots for both variables displayed a large spatial variability of the errors, suggesting the limited capacity of RCMs to properly capture spatial variations of both temperature and precipitation patterns. Regarding precipitation, a similar result was obtained by Mascaro et al. (2018) for the Sardinia region. To find a possible explanation, we decided to investigate possible relationships between the amount of the errors and the cells' mean altitude. In particular, correlation analyses between the elevation and the mean and the standard deviation of the mean annual air temperature and precipitation errors were carried out. Nonetheless, results, here not shown for the sake of brevity, did not provide significant correlations. Turning to seasonal variability, some similarities between mean temperature and precipitation arise in spring, with the ECE-CCLM model looking valuable for both variables. ECE models also perform well in winter but in combination with different RCMs (i.e. HIRH for temperature and RACM for precipitation). In summer, MPI-REMOr2 model is the best option for precipitation but works well also for mean temperature, mainly for Zones 5 and 6. In autumn, MPI-REMOr2 is once again the best performing model but for mean temperature only. Alternatively, MPI-CCLM looks valuable for both mean temperature and precipitation during this season, as also confirmed by the Taylor diagrams (Figs. 4 and 9). Finally, the best models for drought intensity broadly recall those identified for annual precipitation, specifically for ECE-CCLM and Had-RACM.

The skills of CMs in reproducing drought characteristics and variability of precipitation are significantly linked. Drought characteristics, derived through the application of theory of runs, are functions of the departure from the thresholds rather than of the modelled precipitation itself. In other words, although a CM could significantly underestimate or overestimate annual and seasonal precipitation values (i.e. the data in the boxplots in Fig. 6a and 8 may look loosely grouped and the medians very far from 0), still it could provide good performance in terms of drought characteristics simulation if it can reproduce time variability. It is interesting to observe that the distribution of the percentage error of drought intensity (Fig. 11c and f) is, in general, less scattered than that related to the accumulated deficit (Fig. 11b and e); therefore, one can conclude that a partial error compensation occurs when the modelled accumulated deficit is divided by the modelled duration. Despite the differences in the percentage errors, however, there is a general agreement in the identification of the best and, mainly, the worst models, also confirmed by the ranking of the models in reproducing drought intensity and return period of drought events with fixed duration (Fig. 14a and b) both at the annual and the seasonal time scale.

### 5.3 Impact of GCM and RCM choice and different realizations

Overall, no GCM prevails on the others because the RCMs deeply affect the final results. For example, concerning annual precipitation, the simulations relying on the Had GCM provide two high-ranked models (i.e., Had-CCLM and Had-RACM) and a low-ranked model (i.e., Had-RCA4). In the case of precipitation, only one among the GCMs used more than once coherently provides always bad results (IPS).

Concerning the most used RCMs, CCLM seems able to improve performances always with temperature (Fig. 3) and in most cases with precipitation (Fig. 7). Also, RACM usually provides high rankings with precipitation, while lower performances are found with temperature. The five occurrences of RCA4 very seldom provide high rankings with precipitation, as well as the two occurrences of HIRH.

It is of some interest to analyse the behaviour of different realizations of the same CM, which provide insight into the effects of the variability of a multi-member GCM ensemble (von Trentini et al., 2019). In this study, two cases occur, i.e., ECE_RACM and MPI_REMO. Looking at all the box-plots and Taylor diagrams, the two versions of the models behave rather coherently. Nevertheless, because of the variability of the overall model ensemble, usually, they are not ranked in subsequent positions. E.g., considering annual drought ranking and the total area, ECE-RACM is ranked 3[rd] and ECE-RACMr12 17[th], while in the seasonal drought ranking MPI-REMO is ranked 7[th] and MPI-REMOr2 15[th]. This result highlights that, at least to a certain extent, the variability induced by different driving ensemble members is of the same order of the variability given by other GCM-RCM combinations. On the other hand, given the similar performances of the different realizations pointed out by the box-plots and Taylor diagrams, it is confirmed that rather slight differences in models' performance can be found even for distances of 4-5 positions in the rankings.

### 5.4 Overall ranking and comparison with literature

For a final evaluation of the models, an overall ranking criterion was applied. This ranking takes into consideration both the skills of the considered GCM-RCMs models to replicate annual precipitation and temperature variability, as well as drought characteristics. As shown in Fig. 16, the models with the best overall performances, both in the whole case study area and in the six climatically homogeneous zones are those in combination with CCLM RCMs, with the significant exception of Had-RACM, which is ranked 1[st] considering the total area and Zones 2 and 4 for the annul time scale. Generally, the worst models both at the annual and the seasonal scale are Nor-HIRH, IPS-WRF, and CM5-RCA4, although at the seasonal scale also CM5-ALAD and CM5-ALAR have poor performances.

An attempt can be made to compare the results of our ranking exercise with similar studies. Such a comparison is here limited to the Euro-CORDEX climate models for which, indeed, only a few studies do exist. Perhaps the study from Kotlarski et al. (2014) allows the most interesting comparisons for our purposes, being focused on both precipitation and temperature at seasonal and yearly timescales, and covering all areas of Europe, with specific results for the Mediterranean area. Models here denoted as CCLM (CLMCOM-11 in the mentioned study) perform well in reproducing annual temperature and precipitation

in both studies. Differences arise for precipitation in the MAM season, since CCLM models show poor performances according to Kotlarski et al. (2014), in contrast to our findings. Mascaro et al. (2018), whose study is focused on the Sardinia region (Italy), also found that the Had-RACM and ECE-CCLM models perform well in reproducing annual precipitation, while there is no agreement on the CM5-ALAD model. At the seasonal level, ECE-RACMr12, MPI-REMOr2 and MPI-CCLM perform well in both studies in the seasons DJF, JJA, SON respectively, while, in contrast to our results, in the MAM season the ECE-CCLM does not perform well. These differences in the ranking could be partially due to the different observational datasets used, which have found to play a key role in climate model evaluations (Kotlarski et al., 2017).

## 6 Conclusions

In this study we compared the skill of nineteen EURO-CORDEX RCMs at 0.11° (~ 12.5 km) grid spatial resolution in reproducing the annual and seasonal temperature and precipitation regime, as well as several drought features, observed in the period 1971-2000 in a dense network of rain gauges in Sicily and Calabria regions (Southern Italy). From our investigation study a few general and specific conclusions can be drawn. From a general point of view, the model combinations are able to simulate temperature better than precipitation, even though important biases do exist in both variables. Models which are reliable in simulating precipitation may not be the same respect to temperature. This is the case, for instance, of the ECE-RACM model which is in the top ranks for precipitation, while being in the lower ranks for temperature. Models that perform best for precipitation do almost the same for drought features. Differences between the rankings of annual respect to seasonal characteristics do exist, but top-ranking models at the annual scale mostly perform well in the single season, both for precipitation and temperature. Looking more specifically to the models, the Had-RACM, ECE-CCLM, Had-CCLM and ECE-RACM are those that perform best for precipitation and drought, while the CM5-RCA4 and IPS-WRF are those that perform worst. For temperature, models that perform best are MPI-CCLM, MPI-REMO and Had-CCLM, while the worst are CM5-ALAD, ECE-RCA4, ECE-RACM and CM-RCA4. Had-CCLM performs well for both precipitation and temperature, while the CM5-RCA4 performs bad for both.

Results of this study reveal insight on RCMs performances in small-scale regions, which are often targeted by impact studies and have so far received less attention, and provide some guidance to select the best models about the variable and the area under investigation. This is a key issue before addressing projections changes in the evolution of extreme hydro-meteorological events, such as drought characteristics (frequency, duration, and magnitude).

**Data availability.** Ground-based datasets are provided, upon request, by the "Centro Funzionale Multirischi – ARPACAL" (http://www.cfd.calabria.it/ - for Calabria) and the "Osservatorio delle Acque – Regione Sicilia" (www.osservatorioacque.it – for Sicily). Climate data are freely available at the EURO-CORDEX WEBSITE (https://www.euro-cordex.net/).

**Author contribution.** A.S., D.J.P. and B.B. designed the experiments. A.S. and B.B. contributed to sample preparation and preliminary data analysis. D.J.P. and P.N. performed the main computations. A.C. and G.M. supervised the research. All authors discussed the results and contributed to the final manuscript.

**Competing interests.** The paper belongs to the special issue "Recent advances in drought and water scarcity monitoring, modelling and forecasting". Dr. Brunella Bonaccorso, co-author of the paper, is co-editor of the aforementioned special issue. Therefore, she did not assume the role of handling editor for this paper. The other authors declare that they have no conflict of interest.

**Acknowledgements.** The authors thank the "Centro Funzionale Multirischi – ARPACAL" and the "Osservatorio delle Acque – Regione Sicilia" for providing the observed precipitation and temperature data.

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

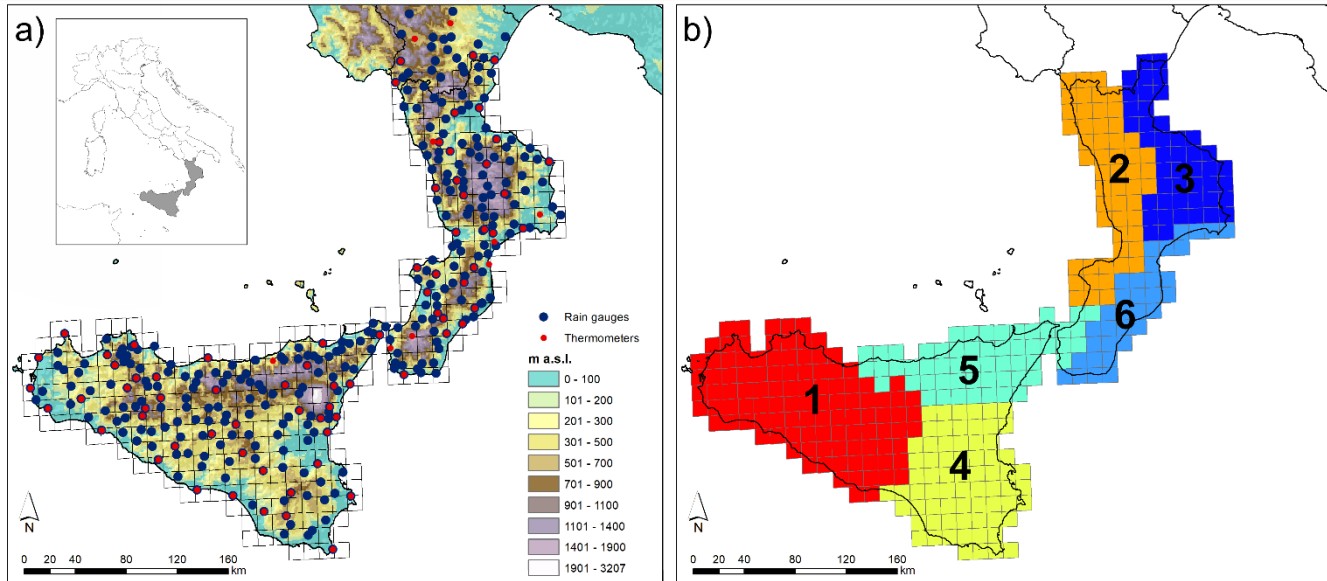

**Figure 1.** a) Study area (Calabria is the southernmost peninsula of Italy and Sicily is the neighbouring island) with the locations of the gauges of the high-density observational network and the CORDEX reference grid; b) the six homogeneous zones identified through PCA.

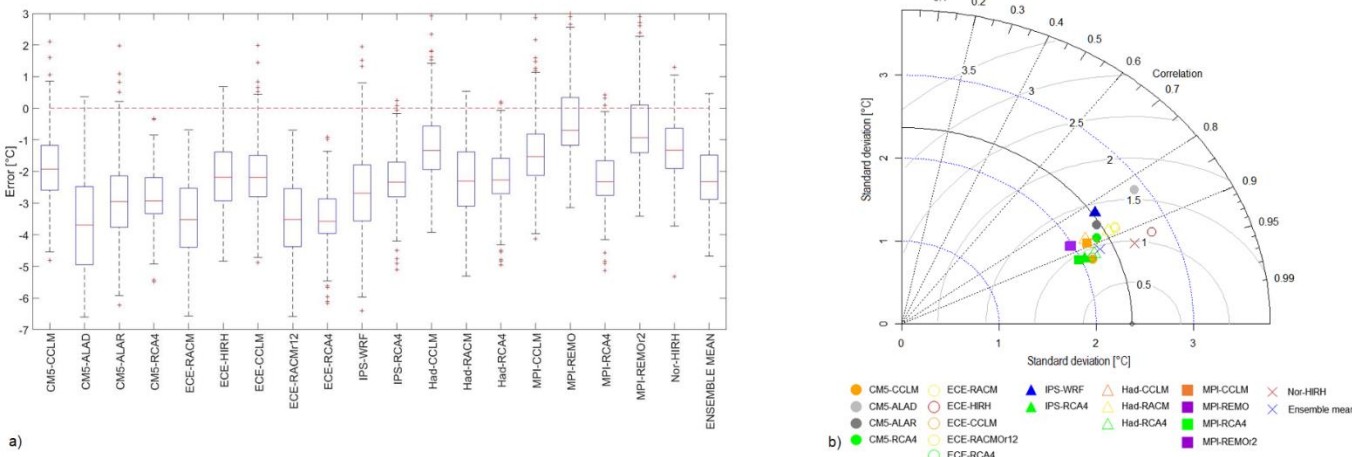

**Figure 2.** a) Box-plots representing the frequency distribution of RCMs errors in mean annual temperature for the whole study area. b) Taylor diagram comparing models performances in reproducing the interannual variability of mean annual temperature for the whole study area.

| | ANNUAL | | | | | | |
|---|---|---|---|---|---|---|---|
| CM5-CCLM | 6 | 7 | 7 | 4 | 6 | 5 | 6 |
| CM5-ALAD | 19 | 18 | 19 | 19 | 17 | 17 | 18 |
| CM5-ALAR | 14 | 15 | 13 | 15 | 13 | 13 | 15 |
| CM5-RCA4 | 15 | 14 | 14 | 12 | 16 | 15 | 13 |
| ECE-RACM | 16 | 16 | 15 | 17 | 14 | 14 | 16 |
| ECE-HIRH | 10 | 10 | 11 | 9 | 10 | 9 | 10 |
| ECE-CCLM | 8 | 8 | 8 | 8 | 7 | 8 | 9 |
| ECE-RACMr12 | 18 | 19 | 18 | 18 | 18 | 18 | 19 |
| ECE-RCA4 | 17 | 17 | 17 | 16 | 19 | 19 | 17 |
| IPS-WRF | 12 | 13 | 12 | 14 | 12 | 6 | 11 |
| IPS-RCA4 | 4 | 5 | 4 | 1 | 4 | 7 | 1 |
| Had-CCLM | 3 | 2 | 3 | 3 | 3 | 4 | 5 |
| Had-RACM | 11 | 11 | 10 | 11 | 9 | 10 | 12 |
| Had-RCA4 | 7 | 6 | 6 | 5 | 8 | 11 | 8 |
| MPI-CCLM | 1 | 3 | 1 | 2 | 1 | 1 | 3 |
| MPI-REMO | 2 | 1 | 2 | 6 | 2 | 2 | 2 |
| MPI-RCA4 | 9 | 9 | 9 | 7 | 11 | 12 | 7 |
| MPI-REMOr2 | 5 | 4 | 5 | 10 | 5 | 3 | 4 |
| Nor-HIRH | 13 | 12 | 16 | 13 | 15 | 16 | 14 |
| | Total Area | Zone 1 | Zone 2 | Zone 3 | Zone 4 | Zone 5 | Zone 6 |

**Figure 3.** RCMs ranking with respect to interannual variability of mean annual temperature, for the entire area and the climatically homogenous zones.

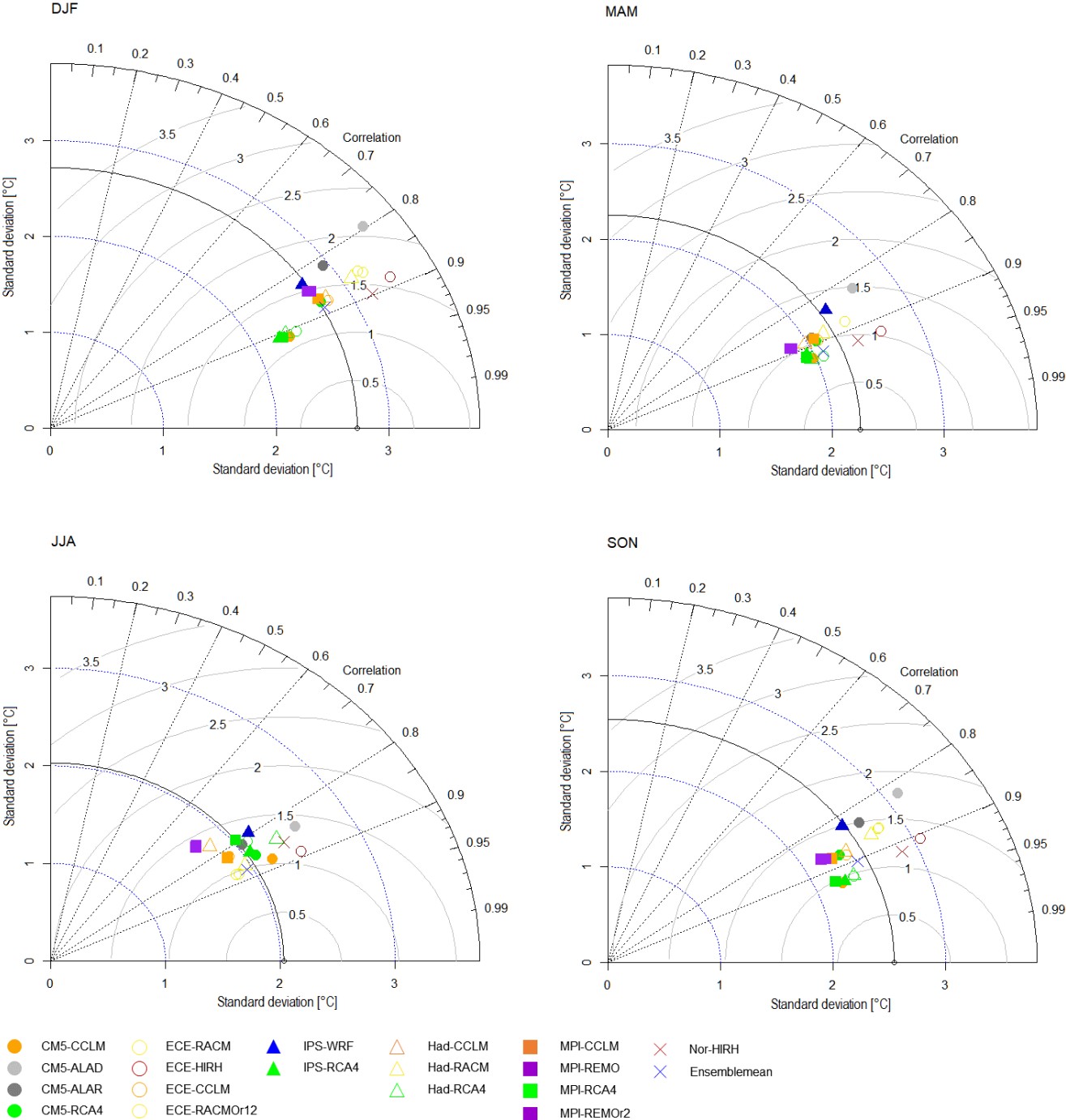

**Figure 4.** Taylor diagram comparing models performances in reproducing the seasonal variability of mean annual temperature for the whole study area.

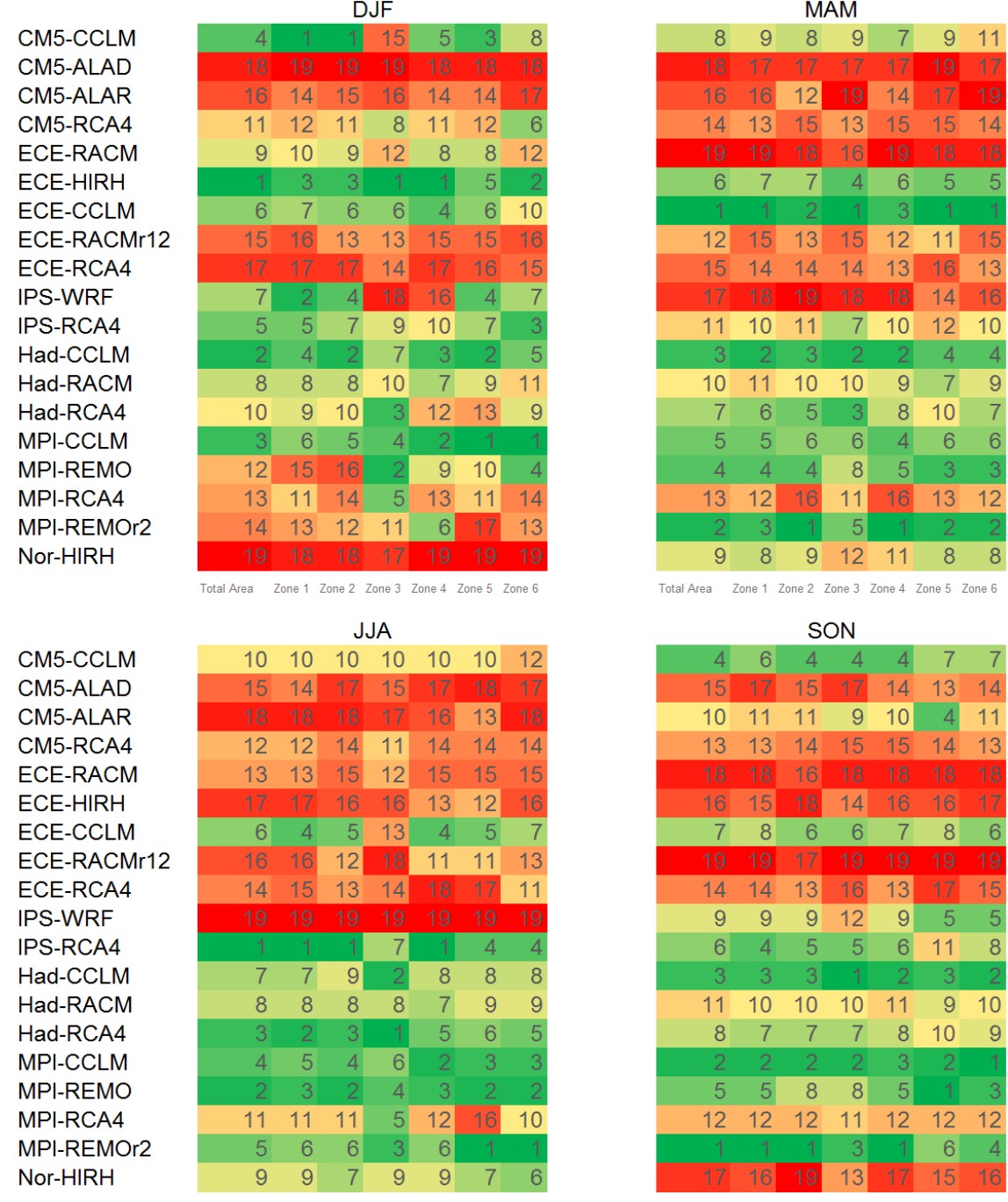

**Figure 5.** RCMs ranking with respect to seasonal variability of mean annual temperature, for the entire area and the climatically homogenous zones.

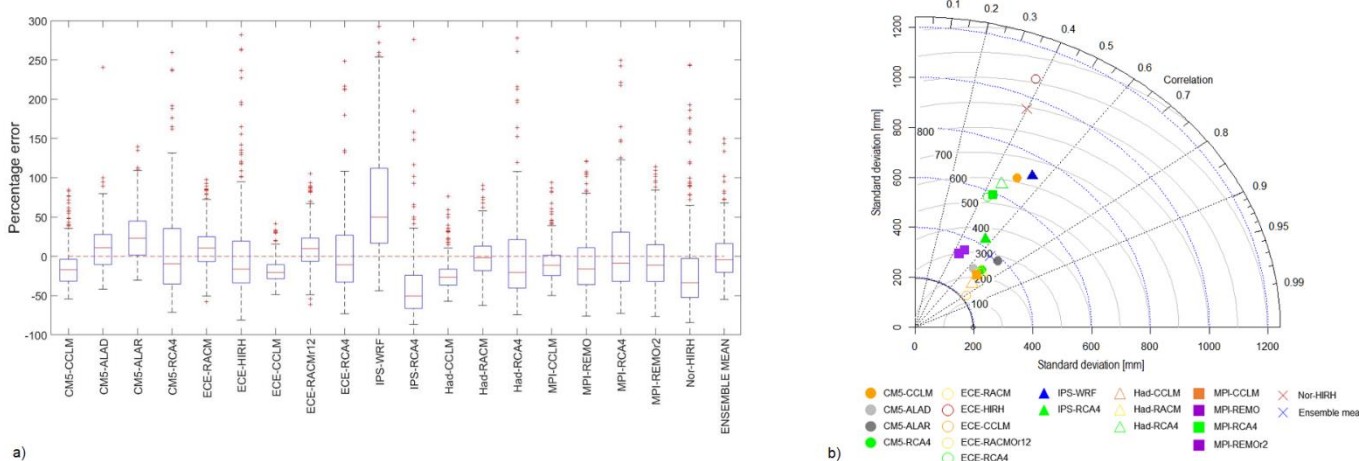

**Figure 6.** a) As Fig. 2a but for annual precipitation. b) As Fig. 2b but for annual precipitation.

ANNUAL

| | Total Area | Zone 1 | Zone 2 | Zone 3 | Zone 4 | Zone 5 | Zone 6 |
|---|---|---|---|---|---|---|---|
| CM5-CCLM | 7 | 2 | 12 | 4 | 6 | 9 | 4 |
| CM5-ALAD | 3 | 5 | 4 | 5 | 11 | 2 | 1 |
| CM5-ALAR | 9 | 13 | 3 | 9 | 4 | 7 | 12 |
| CM5-RCA4 | 17 | 16 | 14 | 16 | 15 | 18 | 19 |
| ECE-RACM | 5 | 3 | 1 | 14 | 8 | 5 | 8 |
| ECE-HIRH | 15 | 11 | 13 | 15 | 18 | 17 | 18 |
| ECE-CCLM | 2 | 4 | 6 | 1 | 1 | 1 | 6 |
| ECE-RACMr12 | 4 | 6 | 5 | 6 | 5 | 4 | 3 |
| ECE-RCA4 | 11 | 9 | 10 | 12 | 14 | 11 | 11 |
| IPS-WRF | 19 | 19 | 19 | 11 | 12 | 19 | 15 |
| IPS-RCA4 | 16 | 18 | 17 | 19 | 17 | 10 | 14 |
| Had-CCLM | 6 | 8 | 7 | 2 | 3 | 6 | 7 |
| Had-RACM | 1 | 1 | 2 | 3 | 2 | 3 | 2 |
| Had-RCA4 | 14 | 12 | 15 | 17 | 16 | 14 | 17 |
| MPI-CCLM | 8 | 7 | 9 | 8 | 7 | 8 | 5 |
| MPI-REMO | 12 | 14 | 16 | 10 | 9 | 15 | 9 |
| MPI-RCA4 | 13 | 10 | 8 | 13 | 13 | 12 | 13 |
| MPI-REMOr2 | 10 | 15 | 11 | 7 | 10 | 13 | 10 |
| Nor-HIRH | 18 | 17 | 18 | 18 | 19 | 16 | 16 |

**Figure 7.** As Fig. 3 but for annual precipitation.

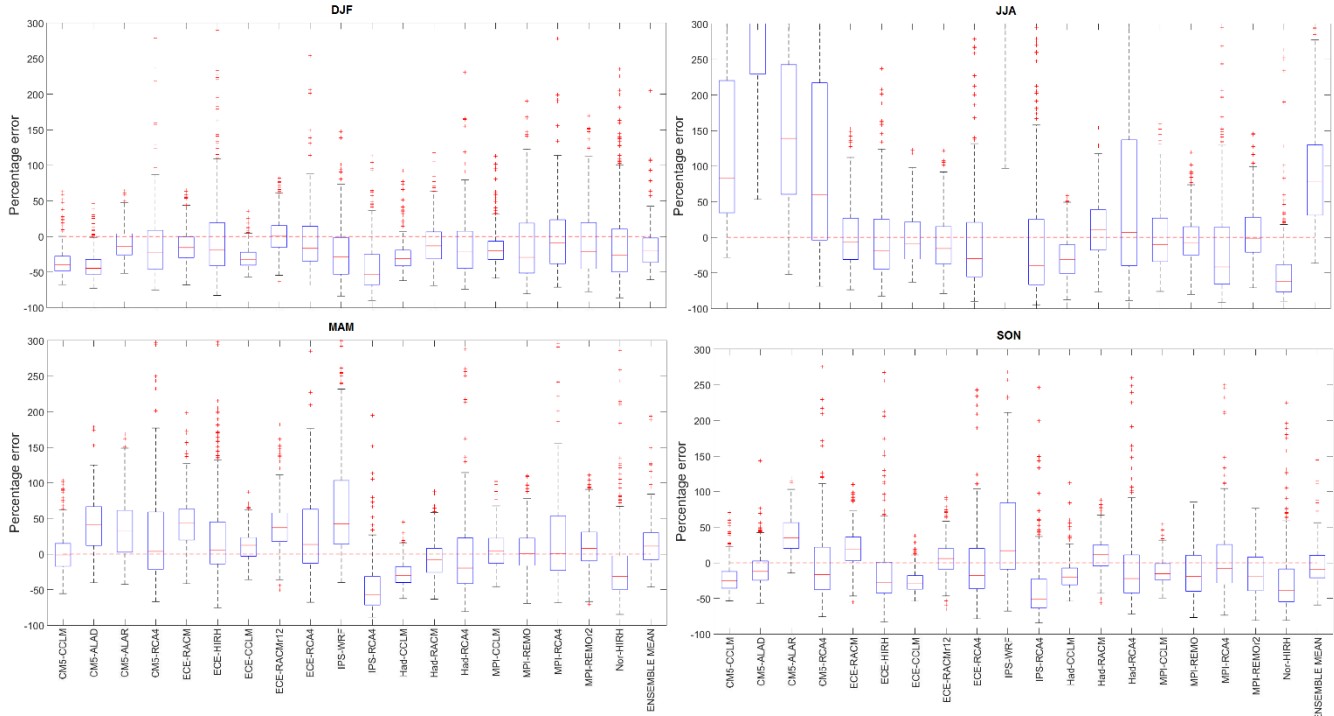

**Figure 8.** Box-plots representing the frequency distribution of RCMs percentage errors in seasonal precipitation for the whole study area.

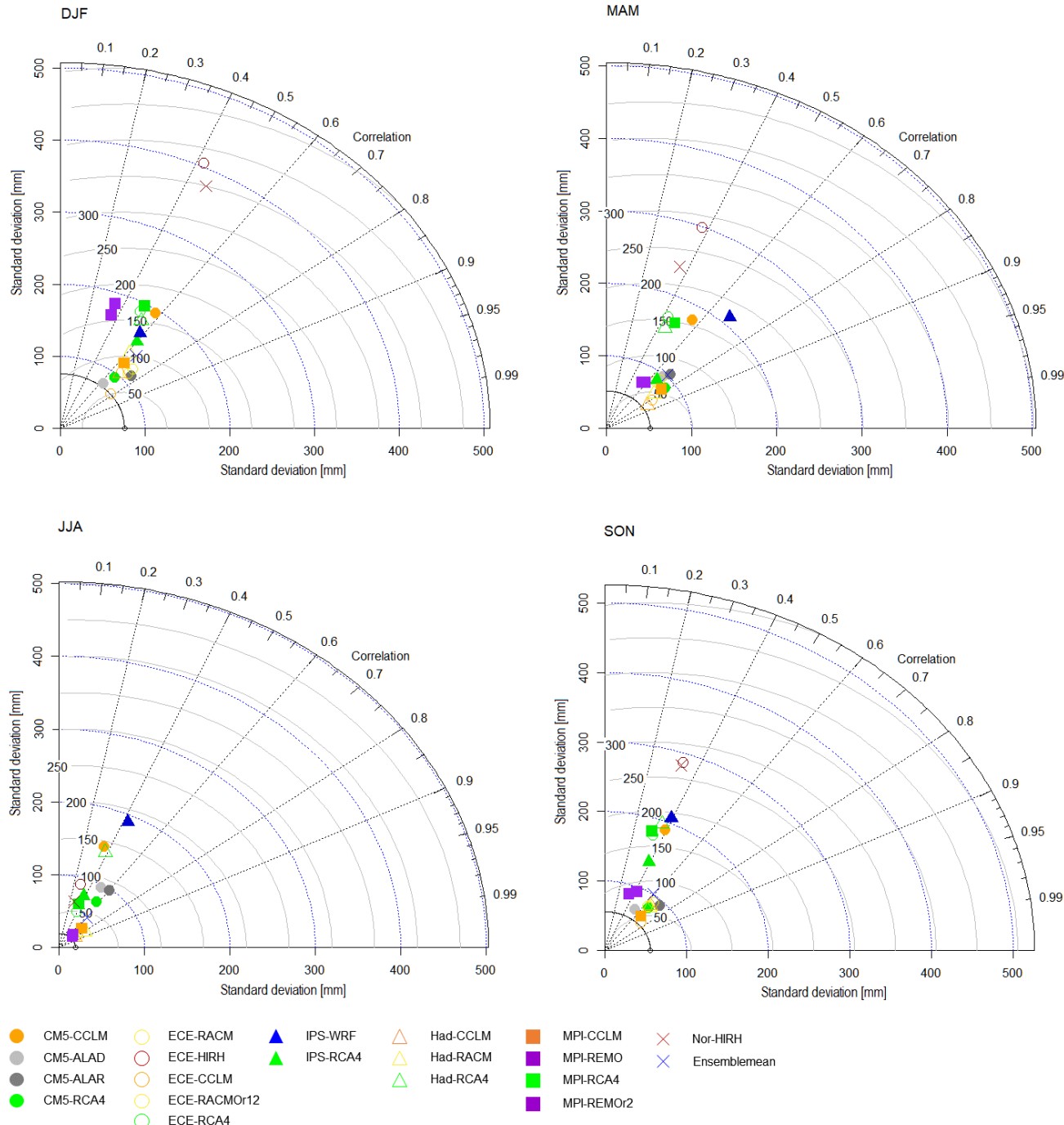

**Figure 9.** As Fig. 4 but for mean annual precipitation.

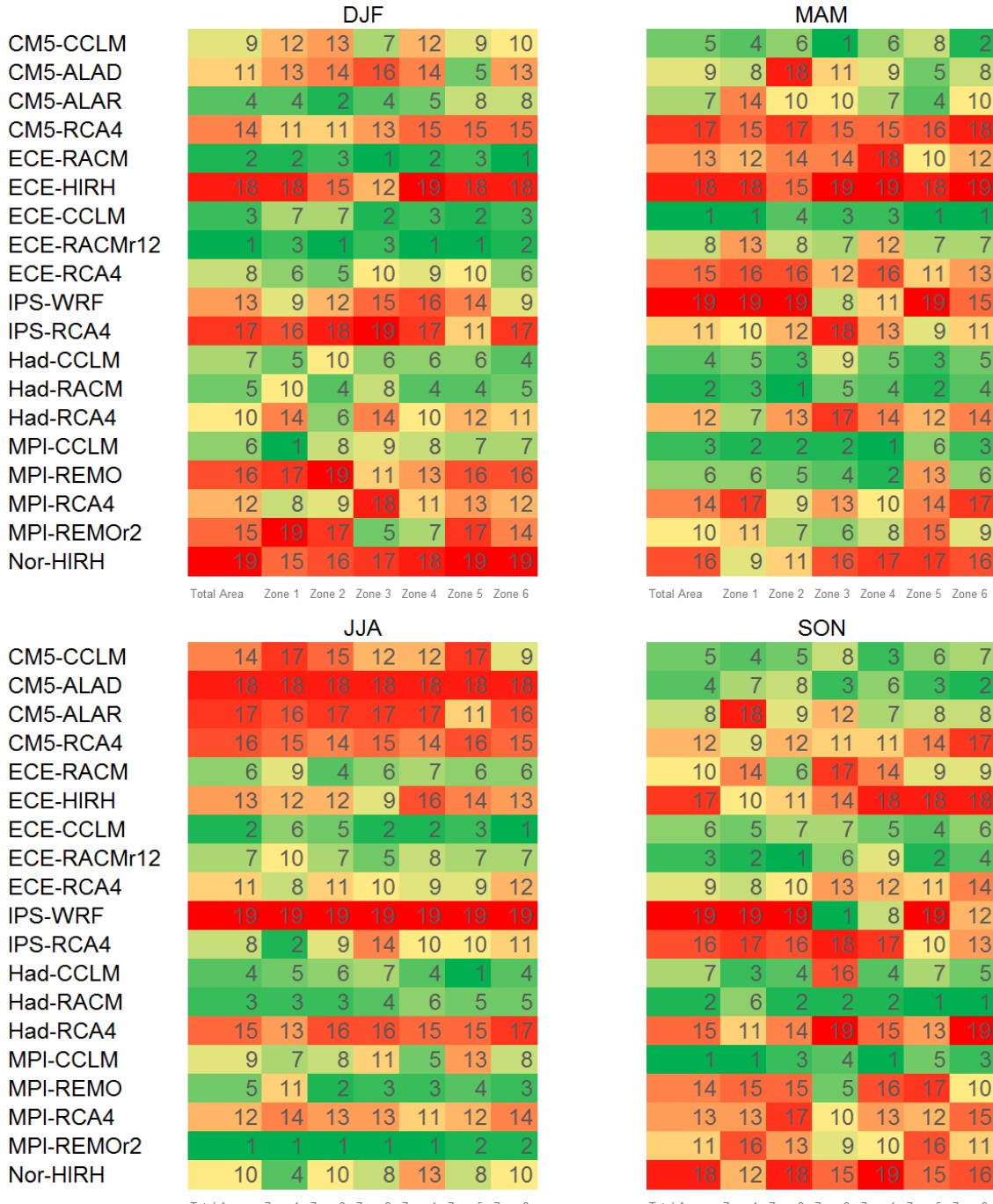

**Figure 10.** As Fig. 5 but for seasonal precipitation.

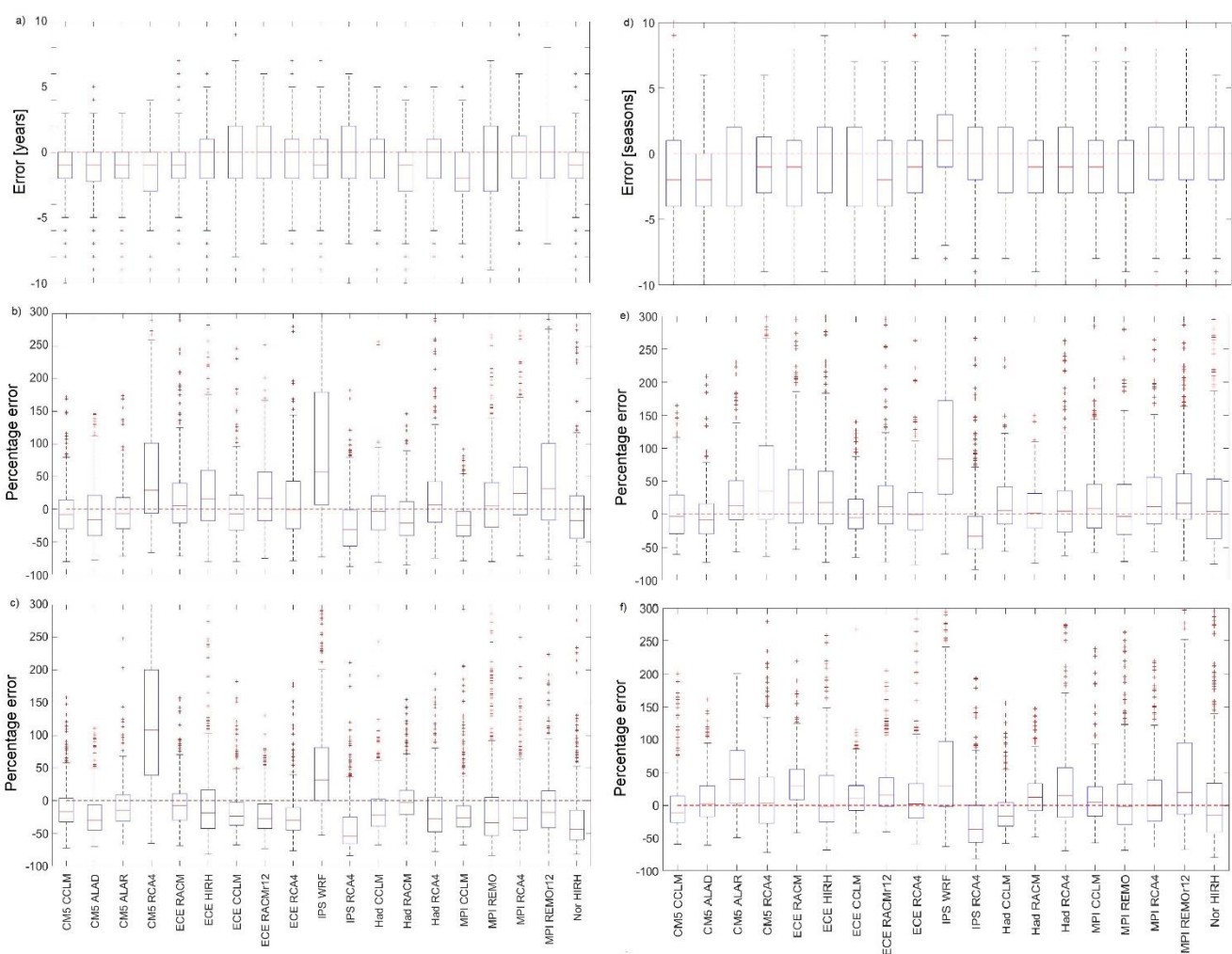

**Figure 11.** Box-plots representing the frequency distribution of RCMs percentage errors in: a) maximum drought duration (annual analysis); b) maximum drought accumulated deficit (annual analysis); c) maximum drought intensity (annual analysis); d) maximum drought duration (seasonal analysis); e) maximum drought accumulated deficit (seasonal analysis); f) maximum drought intensity (seasonal analysis).

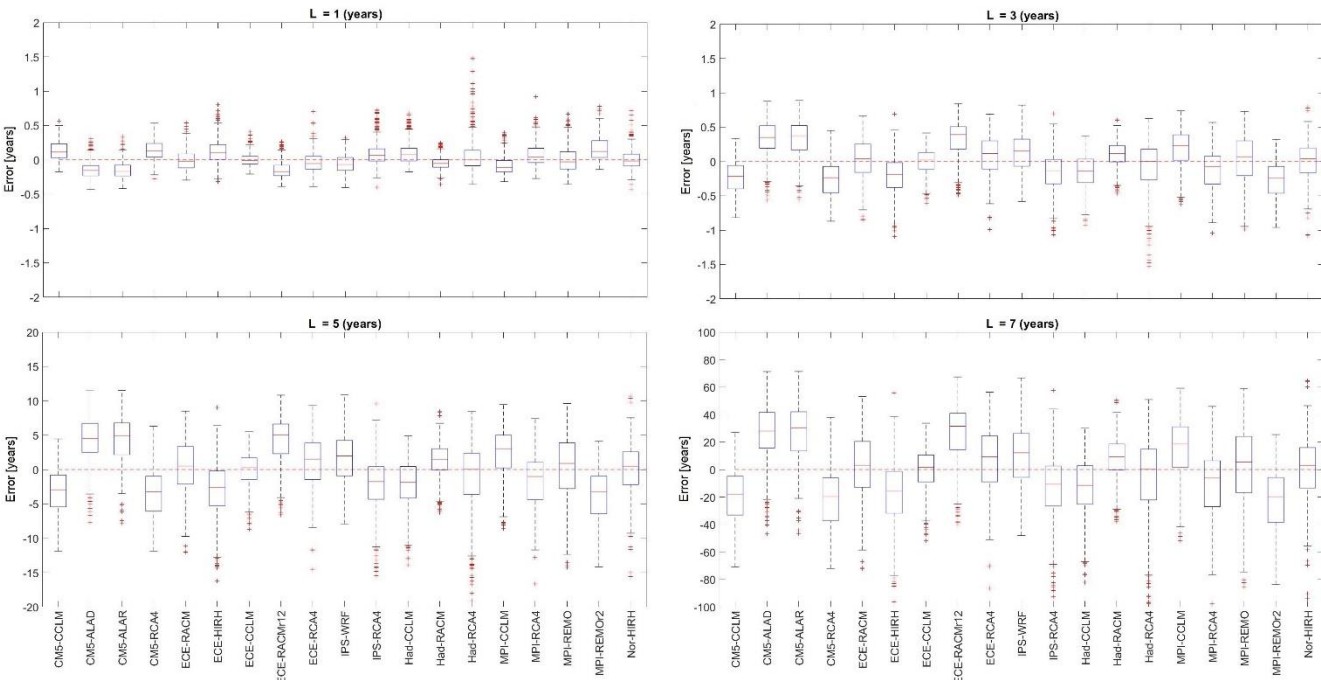

**Figure 12.** Box-plots representing the frequency distribution of RCMs errors in the return period of drought events of duration L equal to 1, 3, 5 and 7 years.

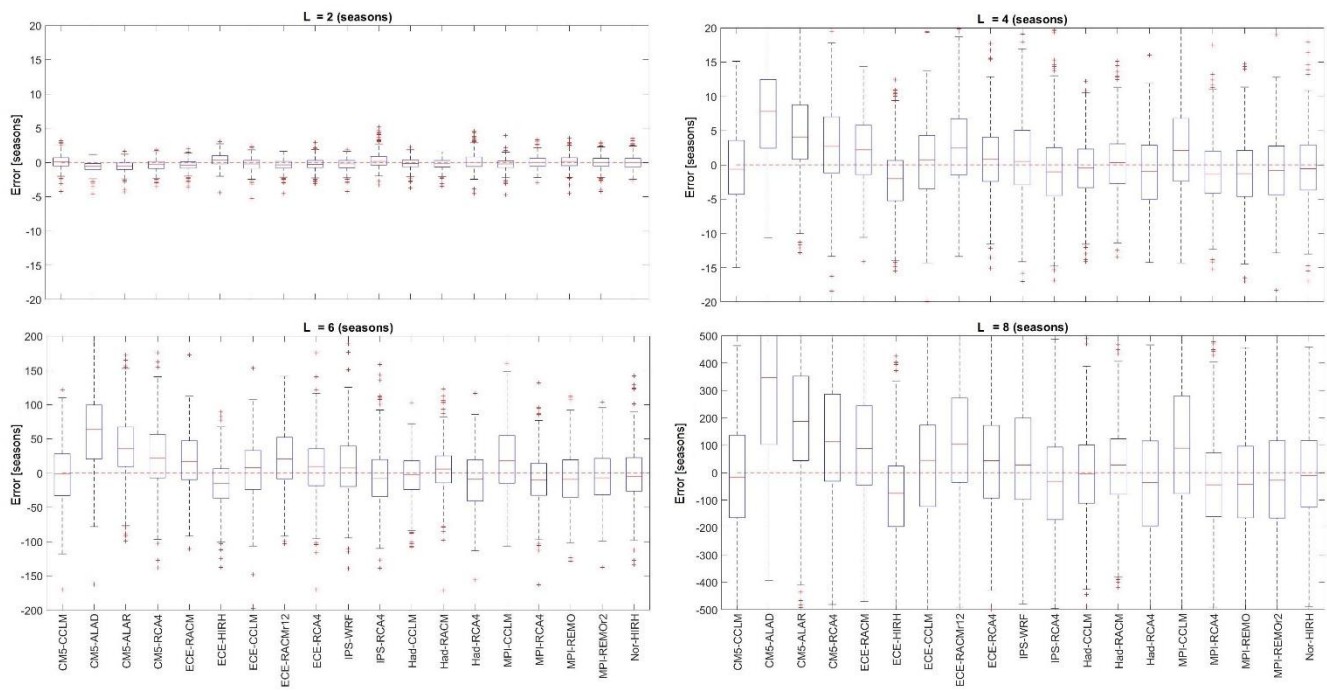

**Figure 13**. Box-plots representing the frequency distribution of RCMs percentage errors in the return period of drought event of duration L equal to 2, 4, 6 and 8 seasons.

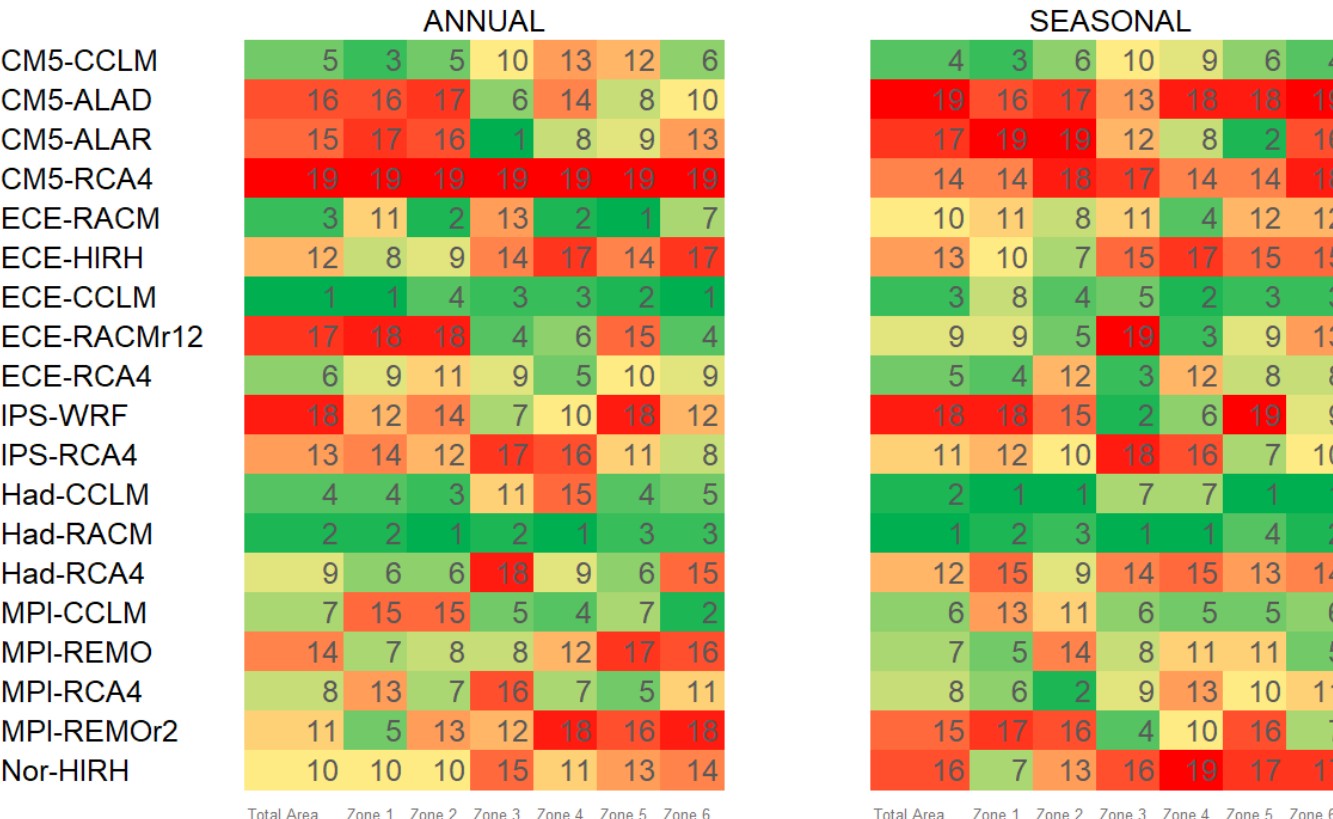

**Figure 14**. RCMs ranking with respect to their ability in reproducing both observed drought maximum intensities and return periods of drought events with duration L=3 years (left) and L=4 seasons (right).

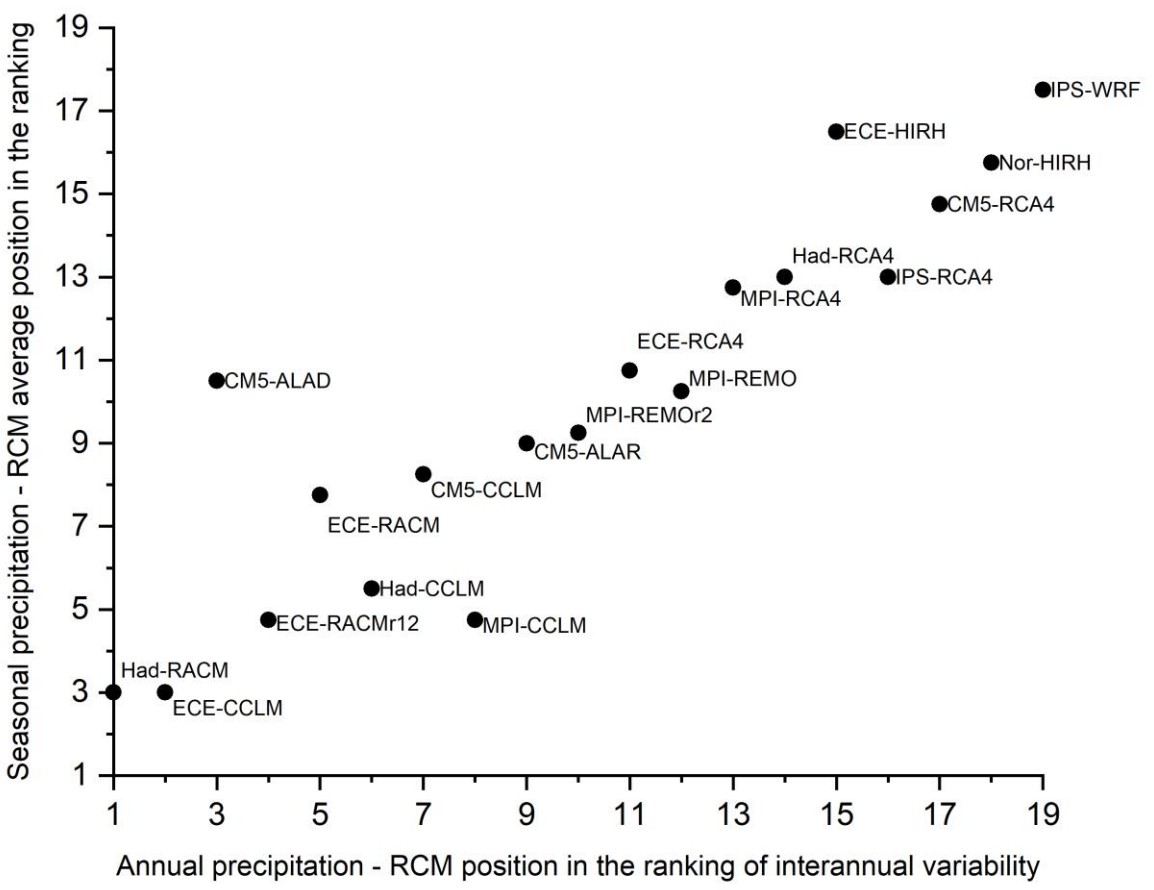

**Figure 15**. Comparison between the RCM position in the ranking of interannual variability of annual precipitation versus the average position in the ranking of seasonal variability of seasonal precipitation. Data concerns the whole study area (Calabria and Sicily).

## ANNUAL

| | Total Area | Zone 1 | Zone 2 | Zone 3 | Zone 4 | Zone 5 | Zone 6 |
|---|---|---|---|---|---|---|---|
| CM5-CCLM | 4 | 3 | 6 | 4 | 7 | 7 | 3 |
| CM5-ALAD | 13 | 15 | 16 | 8 | 16 | 6 | 9 |
| CM5-ALAR | 10 | 14 | 7 | 6 | 6 | 8 | 15 |
| CM5-RCA4 | 19 | 17 | 14 | 19 | 19 | 19 | 19 |
| ECE-RACM | 5 | 8 | 3 | 16 | 4 | 2 | 8 |
| ECE-HIRH | 16 | 11 | 12 | 14 | 17 | 15 | 18 |
| ECE-CCLM | 2 | 1 | 4 | 1 | 2 | 1 | 4 |
| ECE-RACMr12 | 14 | 18 | 18 | 7 | 9 | 10 | 6 |
| ECE-RCA4 | 8 | 13 | 15 | 13 | 15 | 14 | 12 |
| IPS-WRF | 18 | 19 | 19 | 10 | 14 | 16 | 14 |
| IPS-RCA4 | 11 | 12 | 13 | 17 | 13 | 9 | 7 |
| Had-CCLM | 3 | 2 | 2 | 5 | 5 | 3 | 2 |
| Had-RACM | 1 | 4 | 1 | 3 | 1 | 4 | 5 |
| Had-RCA4 | 15 | 9 | 10 | 15 | 12 | 12 | 17 |
| MPI-CCLM | 6 | 7 | 5 | 2 | 3 | 5 | 1 |
| MPI-REMO | 7 | 6 | 8 | 9 | 8 | 17 | 11 |
| MPI-RCA4 | 12 | 10 | 9 | 12 | 10 | 11 | 10 |
| MPI-REMOr2 | 9 | 5 | 11 | 11 | 11 | 13 | 13 |
| Nor-HIRH | 17 | 16 | 17 | 18 | 18 | 18 | 16 |

## SEASONAL

| | Total Area | Zone 1 | Zone 2 | Zone 3 | Zone 4 | Zone 5 | Zone 6 |
|---|---|---|---|---|---|---|---|
| CM5-CCLM | 5 | 5 | 5 | 7 | 7 | 5 | 7 |
| CM5-ALAD | 16 | 17 | 19 | 19 | 17 | 12 | 15 |
| CM5-ALAR | 14 | 18 | 16 | 17 | 9 | 6 | 18 |
| CM5-RCA4 | 18 | 16 | 17 | 15 | 16 | 19 | 19 |
| ECE-RACM | 10 | 15 | 11 | 12 | 10 | 11 | 11 |
| ECE-HIRH | 15 | 14 | 13 | 11 | 18 | 16 | 16 |
| ECE-CCLM | 2 | 3 | 2 | 1 | 2 | 1 | 2 |
| ECE-RACMr12 | 9 | 10 | 6 | 10 | 8 | 8 | 9 |
| ECE-RCA4 | 12 | 12 | 14 | 13 | 13 | 15 | 12 |
| IPS-WRF | 19 | 19 | 18 | 14 | 15 | 17 | 14 |
| IPS-RCA4 | 8 | 6 | 9 | 16 | 12 | 9 | 8 |
| Had-CCLM | 1 | 1 | 1 | 6 | 3 | 2 | 1 |
| Had-RACM | 3 | 4 | 3 | 2 | 4 | 4 | 5 |
| Had-RCA4 | 11 | 8 | 10 | 9 | 11 | 14 | 10 |
| MPI-CCLM | 4 | 2 | 4 | 3 | 1 | 3 | 3 |
| MPI-REMO | 6 | 7 | 8 | 5 | 6 | 7 | 4 |
| MPI-RCA4 | 13 | 13 | 12 | 8 | 14 | 13 | 13 |
| MPI-REMOr2 | 7 | 9 | 7 | 4 | 5 | 10 | 6 |
| Nor-HIRH | 17 | 11 | 15 | 18 | 19 | 18 | 17 |

**Figure 16.** Overall Ranking.

**Table 1.** Intercomparison studies of RCMs' performances within the CORDEX framework

| References | Models | Variables | Region | Main conclusions |
|---|---|---|---|---|
| Schmidli et al. (2007) | 6 statistical downscaling models (SDMs) and 3 RCMs | Daily precipitation | European Alps | SDMs and RCMs tend to have similar biases but differ with respect to interannual variations, with SDMs strongly underestimate the magnitude of the year-to-year variations, mainly in winter. RCMs indicate a strong trend toward drier conditions including longer periods of drought. The SDMs, on the other hand, show mostly non-significant or even opposite changes. |
| Endris et al., (2013) | 10 RCMs from CORDEX Africa domain | Seasonal and annual precipitation | Eastern Africa and 3 sub-regions | RCMs reasonably simulate the main features of the precipitation climatology. However significant biases are detected in individual models depending on sub-region and season. The ensemble mean has better agreement with observation than individual models. |
| Kotlarski et al. (2014) | 9 EURO-CORDEX RCMs | Spatiotemporal patterns of the European climate | Europe | The analysis confirms the ability of RCMs to capture the basic features of the European climate. Seasonally and regionally averaged temperature biases are mostly smaller than 1.5 °C, while precipitation biases are typically located in the ±40% range. |
| Meque and Abiodun (2015) | 10 RCMs from CORDEX Africa domain | Link between El Niño Southern Oscillation (ENSO) and Southern African droughts expressed by the Standardized Precipitation and Evapotranspiration Index (SPEI) | Southern Africa | ARPEGE model shows the best simulation, while CRCM shows the worst. |
| Mascaro et al. (2015) | 6 RCMs driven by 10 GCMs from CORDEX Africa domain | Properties of the hydrological cycle | Niger River basin (West Africa) | Most RCMs overestimate (order of +10% to +400%, depending on model and subbasin) the mean annual difference between precipitation (P) and evaporation (E), |
| Wu et al. (2016) | 4 RCMs from RMIP Project and their regional multi-model ensemble, and their driving GCM ECHAM5 | Summer extreme precipitation | East Asia | All models can adequately reproduce the spatial distribution of extremely heavy precipitation. However, they do not perform well in simulating summer consecutive dry days. The ensemble average of multi-RCMs substantially improve model capability to simulate summer precipitation in both total and extreme categories when compared to each individual RCM. |
| Park et al. (2016) | 5 RCMs form the CORDEX East Asia domain | Climatology of summer extremes (seasonal maxima of daily mean temperature and precipitation) | East Asia | RCMs show systematic bias patterns in both seasonal means and extremes. The models simulate temperature means more accurately compared to extremes because of the higher spatial correlation, whereas precipitation extremes are simulated better than their means because of the higher spatial variability. |

**Table 1.** Continues

| References | Models | Variables | Region | Main conclusions |
|---|---|---|---|---|
| Smiatek et al. (2016) | 13 EURO-CORDEX RCMs | mean temperature and precipitation, frequency distribution of precipitation intensity, maximum number of consecutive dry days | Greater Alpine Region (GAR) | Though the models reproduce spatial seasonal precipitation patterns, the seasonal mean temperature is underestimated (from -0.8 °C to -1.9 °C) and mean precipitation is overestimated (from +14.8% in summer to +41.5% in winter). Larger errors are found for further statistics and various GAR sub-regions. |
| Diasso and Abiodun (2017) | 10 RCMs from CORDEX Africa domain | Drought characteristics evaluated through 4 Principal Components of the SPEI | West Africa | Only two models (REMO and CNRM) reproduce all the four drought modes. REMO and WRF give the best simulation of the seasonal variation of the drought mode over the Sahel in March-May and June-August seasons, while CNRM gives the best simulation of seasonal variation in the drought pattern over the Savanna. |
| Um et al. (2017) | 4 RCMs from CORDEX East Asia domain, their ensemble mean and a driving GCM | Drought characteristics based on the SPEI | East Asia | Drought severity diverges markedly among the RCMs. Estimates of drought spatial extent are generally accurate in wet regions but inaccurate in dry regions. In general, the spatial extents of the droughts diverge among the RCMs, and the models fail to accurately capture droughts with large spatial scales. |
| Foley and Kelman (2018) | 7 EURO-CORDEX RCMs and 5 driving GCMs | Several precipitation indices (accumulated precipitation amount, mean daily precipitation amount, max 1-day and 5-day precipitation amounts, simple daily intensity, number of heavy and very heavy precipitation days) | Scottish islands | While no models perform skilfully across all the metrics studied, some models capture aspects of the precipitation climate at each location particularly well. |
| Adeniyi and Dilau (2018) | 10 RCMs from CORDEX Africa domain | Precipitation, temperature and drought | West Africa | ARPEGE has the highest skill at Guinea Coast, while PRECIS is the most skilful over Savannah and RCA over the Sahel. |
| Senatore et al. (2019) | 8 RCMs from CORDEX South Asia domain | Annual and seasonal precipitation and temperature | Iran and 6 sub-regions | No model is significantly better than others for every season and zone. Some enhancements are obtained by a weighting approach to take into account useful information from every RCM in the sub-zones. More reliable models show a strong precipitation decrease. |
| Di Virgilio et al. (2019) | 6 RCMS from CORDEX Australasia domain | Near-surface max and min temperature and precipitation at annual, seasonal, and daily time scales | Australia | All RCMs showed widespread, statistically significant cold biases in maximum temperature and overestimated precipitation, especially over Australia's populous eastern seaboard. |

**Table 2.** List of GCMs, together with the abbreviations used in this paper, included at least once in the EURO-CORDEX ensemble

| Model name | Abbreviation | Reference | Institution |
| --- | --- | --- | --- |
| CNRM-CERFACS-CNRM-CM5 | CM5 | Voldoire et al. (2013) | Centre National de Recherches Météorologiques |
| ICHEC-EC-EARTH | ECE | Hazeleger et al. (2010) | Irish Centre for High-End Computing<br><br>EC-Earth Consortium, Europe |
| IPSL-IPSL-CM5A-MR | IPS | Dufresne et al. (82013) | Institut Pierre Simon Laplace |
| MOHC-HadGEM2-ES | Had | Collins et al. (2011) | Met Office Hadley Centre |
| MPI-M-MPI-ESM-LR | MPI | Giorgetta et al. (2013) | Max-Planck-Institute für Meteorologie |
| NCC-NorESM1-M | Nor | Bentsen et al. (2013), Iversen et al. (2013) | Norwegian Earth System Model |

 **Table 3.** List of RCMs, together with the abbreviations used in this paper, included at least once in the EURO-CORDEX ensemble

| Model name | Abbreviation | Reference | Institution |
|---|---|---|---|
| CNRM-ALADIN53 | ALAD | Colin et al. (2010) | Météo-France / Centre National de Recherches Météorologiques |
| RMIB-UGent-ALARO-0 | ALAR | De Troch et al. (2013) | Royal Meteorological Institute of Belgium and Ghent University |
| CLMcom-CCLM4-8-17 | CCLM | Baldauf et al. (2011), Rockel et al. (2008)  Baldauf et al. (2011), Rockel et al. (2008) | Climate Limited-area Modelling Community (CLM-Community) |
| DMI-HIRHAM5 | HIRH | Christensen et al. (2007) | Danish Meteorological Institute |
| KNMI-RACMO22E | RACM | van Meijgaard et al. (2008) | Royal Netherlands Meteorological Institute, De Bilt, The Netherlands |
| SMHI-RCA4 | RCA4 | Strandberg et al. (2014) | Swedish Meteorological and Hydrological Institute, Rossby Centre |
| MPI-CSC-REMO2009 | REMO | Teichmann et al. (2013) | Helmholtz-Zentrum Geesthacht, Climate Service Center, Max Planck Institute for Meteorology |
| IPSL-INERIS-WRF331F | WRF3 | - | Institut Pierre-Simon Laplace and French National Institute for Industrial Environment and Risks (Ineris) |

**Table 4.** List and acronyms of climate models (GCM-RCM combinations) included at least once in the EURO-CORDEX ensemble. The asterisk * means that two versions of the GCM-RCM combination are available

| | CNRM-CERFACS-CNRM-CM5 | ICHEC-EC-EARTH | IPSL-IPSL-CM5A-MR | MOHC-HadGEM2-ES | MPI-M-MPI-ESM-LR | NCC-NorESM1-M |
|---|---|---|---|---|---|---|
| CNRM-ALADIN53 | CM5-ALAD | - | - | - | - | - |
| RMIB-UGent-ALARO-0 | CM5-ALAR | - | - | - | - | - |
| CLMcom-CCLM4-8-17 | CM5-CCLM | ECE-CCLM | - | Had-CCLM | MPI-CCLM | - |
| DMI-HIRHAM5 | - | ECE-HIRH | - | - | - | Nor-HIRH |
| KNMI-RACMO22E | - | ECE-RACM* | - | Had-RACM | - | - |
| SMHI-RCA4 | CM5-RCA4 | ECE-RCA4 | IPS-RCA4 | Had-RCA4 | MPI-RCA4 | - |
| MPI-CSC-REMO2009 | - | - | - | - | MPI-REMO* | - |
| IPSL-INERIS-WRF331F | - | - | IPS-WRF | - | - | - |

**Table 5.** Summary of the statistics involved in the ranking process. Statistics with subscript $_0$ refer to observed values.

| Property | Statistics $k$ | Error $E_{k,m}(j)$ |
|---|---|---|
| Seasonal variability | Seasonal mean | $\lvert\mu_0\big(X\tau(j)\big) - \mu_m\big(X_\tau(j)\big)\rvert$ |
| | Seasonal standard deviation | $\lvert\sigma_0\big(X\tau(j)\big) - \sigma_m\big(X_\tau(j)\big)\rvert$ |
| Interannual variability | Annual mean | $\lvert\mu_0\big(X(j)\big) - \mu_m\big(X(j)\big)\rvert$ |
| | Annual standard deviation | $\lvert\sigma_0\big(X(j)\big) - \sigma_m\big(X(j)\big)\rvert$ |
| Drought characteristics | Maximum drought duration | $\lvert L_{max,0}(j) - L_{max,m}(j)\rvert$ |
| | Maximum drought accumulated deficit | $\lvert D_{max,0}(j) - D_{max,m}(j)\rvert$ |
| | Maximum drought intensity | $\lvert I_{\max,0}(j) - I_{\max,m}(j)\rvert$ |
| | Return period | $\lvert T_{r,0}(j) - T_{r,m}(j)\rvert$ |

**Table 6.** Best performing RCMs according to the ranking at the annual and seasonal scale.

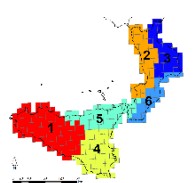

| | | Whole area | Zone 1 | Zone 2 | Zone 3 | Zone 4 | Zone 5 | Zone 6 |
|---|---|---|---|---|---|---|---|---|
| T interannual variability | | MPI-REMO | MPI-REMO | MPI-CCLM | IPS-RCA4 | MPI-CCLM | MPI-CCLM | IPS-RCA4 |
| | | MPI-CCLM | Had-CCLM | MPI-REMO | MPI-CCLM | MPI-REMO | MPI-REMO | MPI-REMO |
| | | Had-CCLM | MPI-CCLM | Had-CCLM | Had-CCLM | Had-CCLM | Had-CCLM | MPI-CCLM |
| T seasonal variability | DJF | ECE-HIRH | CM5-CCLM | CM5-CCLM | ECE-HIRH | ECE-HIRH | MPI-CCLM | MPI-CCLM |
| | MAM | ECE-CCLM | ECE-CCLM | MPI-REMOr2 | ECE-CCLM | MPI-REMOr2 | ECE-CCLM | ECE-CCLM |
| | JJA | IPS-RCA4 | IPS-RCA4 | IPS-RAC4 | Had-RCA4 | IPS-RCA4 | MPI-REMOr2 | MPI-REMOr2 |
| | SON | MPI-REMOr2 | MPI-REMOr2 | MPI-REMOr2 | Had-CCLM | MPI-REMOr2 | MPI-REMOr-12 | MPI-CCLM |
| P interannual variability | | Had-RACM | Had-RACM | ECE-RACM | ECE-CCLM | ECE-CCLM | ECE-CCLM | CM5-ALAD |
| | | ECE-CCLM | CM5-CCLM | Had-RACM | Had-CCLM | Had-RACM | CM5-ALAD | Had-RACM |
| | | CM5-ALAD | CM5-ALAD | CM5-ALAR | Had-RACM | Had-CCLM | Had-RACM | ECE-RACMr12 |
| P seasonal variability | DJF | ECE-RACMr12 | MPI-CCLM | ECE-RACMr12 | ECE-RACM | ECE-RACMr12 | ECE-RACMr12 | ECE-RACM |
| | MAM | ECE-CCLM | ECE-CCLM | Had-RACM | CM5-CCLM | MPI-CCLM | ECE-CCLM | ECE-CCLM |
| | JJA | MPI-REMOr2 | MPI-REMOr2 | MPI-REMOr2 | MPI-REMOr2 | MPI-REMOr2 | Had-CCLM | ECE-CCLM |
| | SON | MPI-CCLM | MPI-CCLM | ECE-RACMr12 | IPS-WRF | MPI-CCLM | Had-RACM | Had-RACM |
| I + $T_y$(L=3 years) (annual scale) | | ECE-CCLM | ECE-CCLM | Had-RACM | CM5-ALAR | Had-RACM | ECE-RACM | ECE-CCLM |
| | | Had-RACM | Had-RACM | ECE-RACM | Had-RACM | ECE-RACM | ECE-CCLM | MPI-CCLM |
| | | ECE-RACM | CM5-CCLM | Had-CCLM | ECE-CCLM | ECE-CCLM | Had-RACM | Had-RACM |
| I+$T_s$(L=4 seasons) (seasonal scale) | | Had-RACM | Had-CCLM | Had-CCLM | Had-RACM | Had-RACM | Had-CCLM | Had-CCLM |
| | | Had-CCLM | Had-RACM | MPI-RCA4 | IPS-WRF | ECE-CCLM | CM5-ALAR | Had-RACM |
| | | ECE-CCLM | CM5-CCLM | Had-RACM | ECE-RCA4 | ECE-RACMr12 | ECE-CCLM | ECE-CCLM |