# Peer review of "Towards a reliable assessment of climate change impact on droughts in Southern Italy: Evaluation of EURO-CORDEX historical simulations by high-quality observational datasets"

_Natural Hazards and Earth System Sciences, 2020_

## Referee Comment (RC1) · Anonymous Referee #1 · 23 Apr 2020

This paper quantifies the performances of several combinations of regional climate models (RCMs) driven by diffenet general circulation models (GCMs) in two regions of southern Italy. The GCM-RCM combinations are part of the Coordinated Regional Climate Downscaling Experiment (CORDEX) initiative in the European domain (EURO-CORDEX). Performances are evaluated in the ability to capture the spatial variability of mean annual and seasonal precipitation (P) and temperature (T), as well as three drought metrics derived by applying the run theory.

I enjoyed reading this paper, which is well written; presents rigorous analyses; critically

discusses the results; and has practical utility for impact studies in the study regions, since it provides a list of best GCM-RCM combinations. I recommend its publication and I only have a few minor requests and suggestions.

1) The authors should provide more details on how they applied the principal component analysis (PCA): • Was it applied on monthly or annual P? • The PCA returns spatial patterns that explain most of the P variability. How did the authors derive the subregions within these spatial patterns? This should be clarified.

2) When ranking the models based on performance in reproducing annual P, the authors find that nine models have similar error metrics. Have they tried to compute the mean rank of each model across the zones and even across the five time scales considered (annual and the four seasons)? There may be some models that are consistently in the top (lowest ranks) and these should be mentioned.

3) Line 45: Extremes occur everywhere. I suggest changing to "...occurrence of particularly intense extreme events, ...". If this is what the authors mean, a reference is also needed.

4) Line 55: CMIP5 has been already defined; just use the acronym.

5) Line 61: I suggest adding "historical" before simulations.

6) Line 326: it should be "show".

---

## Referee Comment (RC2) · Anonymous Referee #2 · 7 May 2020

The paper is well written, the state-of-art well described in the introduction, and the methodology used in this study are clear and can be easily understood from the paper. Overall, the quality of this paper is good, but to be honest I do not think this should be a paper. I mean, I see it more as a technical report or, even better, as the preliminary part of a wider study, maybe from the selection of the best models to dedicated projections of hazard and impacts. I am aware that other studies dealing on the evaluations of newest GCM-RCM simulations do exist, focusing on small regions, e.g. the one cited by authors about Sardinia, but I feel that this is not a research paper, but a (very well per-

formed) study on the performance of models on a test region. Thus, I am questioning myself: once the authors have decided that one combination of GCMs-RCMs performs better than the others, for each quantity analized (precipitation, temperature, drought), time scale (annual, seasonal), sub-region (3 for Sicily, 3 for Calabria).. what shall the reader do with this information inserted in a scientific paper? The region is very small, so - as the authors say (see lines 452-453) - the choice of the best model depends on many factors, making this piece of work not conclusive. What shall be really of interest is what the authors plan for further analyses (Lines 454-456). I also have another major point about the possible publication: the paper is not about droughts. Drought is just slightly touched and with very basic metrics, far from the current standard in drought-relatd analyses, so my final verdict is to reject this submission.

However, I see that authors made great efforts, so they might consider to rethink about the paper and try to resubmit, but I would definitely remove the word droughts from title.

Specific comments: - Why not using also Med-Cordex? - Are the Euro-CORDEX bias-adjusted? Why not using the bias-adjusted runs? - I'd like to see more details on the station data, which could be potentially one of the most interesting parts of the study; - Don't include equations in the core manuscript, move them all to supplementary materials; - Drought part is very poor. Why not using, at least, the SPI and the SPEI? Also the choice of quantities related to drought are not enough to justify the publication, I'd expect a lot more (frequency of events, intensity, severity, return periods, spatial aggregation, etc.) especially on monthly basis (not annual); - Some conclusions are exactly what one might expect: precipitation is modelled worse than temperature, drought (as computed in this study) is similar to precipitation, RCMs deeply affect the results more than GCMs.

---

## Author Comment (AC1) · 22 Jun 2020

**RC:** This paper quantifies the performances of several combinations of regional climate models (RCMs) driven by different general circulation models (GCMs) in two regions of southern Italy. The GCM-RCM combinations are part of the Coordinated Regional Climate Downscaling Experiment (CORDEX) initiative in the European domain (EURO- CORDEX). Performances are evaluated in the ability to capture the spatial variability of mean annual and seasonal precipitation (P) and temperature (T), as well as three drought metrics derived by applying the run theory. I enjoyed reading this paper, which is well written; presents rigorous analyses; critically discusses the results; and has practical utility for impact studies in the study regions, since it provides a list of best GCM-RCM combinations. I recommend its publication and I only have a few minor requests and suggestions.

*AC:   Thanks for your positive comments.*

**RC:** The authors should provide more details on how they applied the principal component analysis (PCA): Was it applied on monthly or annual P? The PCA returns spatial patterns that explain most of the P variability. How did the authors derive the subregions within these spatial patterns? This should be clarified.

*AC:   Thank you for this comment, which allowed us to improve the description of the PCA. At this regard, we will add further information to the manuscript as follows:*

[revised manuscript text omitted]

**RC:** When ranking the models based on performance in reproducing annual P, the authors find that nine models have similar error metrics. Have they tried to compute the mean rank of each model across the zones and even across the five time scales considered (annual and the four seasons)? There may be some models that are consistently in the top (lowest ranks) and these should be mentioned.

*AC: Thank you for this useful suggestion. Indeed, this aspect was already partially investigated and some details are drawn in LL379-384 of the manuscript, where we pointed out that the models Had-RACM, ECE-CCLM and Had-CCLM have the overall better performances at both annual and seasonal time scales. Thus, we propose to enlargen and make clearer the discussion in the revised version of the manuscript. We may also add the figure below, which highlights deviations in the performances of some models (e.g., CM5-ALAD), considering both the annual scale and the average behaviour at the seasonal scale (the higher the deviation, the higher the distance from the bisector).*

[Figure]

*Fig. R1. Comparison between the RCM position in the ranking of interannual variability of annual precipitation versus the average position in the ranking of seasonal variability of seasonal precipitation. Data concerns the whole study area (Calabria and Sicily).*

**RC:** Line 45: Extremes occur everywhere. I suggest changing to "... occurrence of particularly intense extreme events, ...". If this is what the authors mean, a reference is also needed.

**AC:** *That's correct, this is in fact what we meant. The sentence will be changed according to the suggestion and proper references will be added (concerning droughts: Bonaccorso et al., 2013; Bonaccorso et al., 2015a and 2015b; concerning floods: Llasat et al., 2016; Senatore et al., 2020)*

**RC:** Line 55: CMIP5 has been already defined; just use the acronym.

**AC:** *We will revise accordingly.*

**RC:** Line 61: I suggest adding "historical" before simulations.

**AC:** *We will revise accordingly.*

**RC:**   Line 326: it should be "show".

*AC:*   *We will revise accordingly.*

*References to be added in the new version of the manuscript:*

*Bonaccorso, B., Peres D.J., Cancelliere A., Rossi G. (2013). Large Scale Probabilistic Drought Characterization Over Europe. Water Resources Management, 27 (6), pp. 1675-1692, ISSN: 0920-4741, DOI: 10.1007/s11269-012-0177-z.*

*Bonaccorso, B., Peres, D.J., Castano, A., Cancelliere, A. (2015a). SPI-Based Probabilistic Analysis of Drought Areal Extent in Sicily. Water Resources Management, Volume 29(2), pp. 459-470, ISSN: 09204741, DOI: 10.1007/s11269-014-0673-4.*

*Bonaccorso., B., Cancelliere, A., Rossi, G. (2015b). Probabilistic forecasting of drought class transitions in Sicily (Italy) using Standardized Precipitation Index and North Atlantic Oscillation Index. Journal of Hydrology, Volume 526, pp. 136-150, ISSN: 00221694, DOI: 10.1016/j.jhydrol.2015.01.070.*

*Llasat, M.C., Marcos, R., Turco, M., Gilabert, J., and Llasat-Botija, M.: Trends in flash flood events versus convective precipitation in the Mediterranean region: The case of Catalonia, J. Hydrol., 541, 24-37, 10.1016/j.jhydrol.2016.05.040, 2016.*

*Senatore, A., Furnari, L., and Mendicino, G.: Impact of high-resolution sea surface temperature representation on the forecast of small Mediterranean catchments' hydrological responses to heavy precipitation, Hydrol. Earth Syst. Sci., 24, 269–291, https://doi.org/10.5194/hess-24-269-2020, 2020.*

---

## Author Comment (AC2) · 23 Jun 2020

RC:     The paper is well written, the state-of-art well described in the introduction, and the methodology used in this study are clear and can be easily understood from the paper. Overall, the quality of this paper is good, but to be honest I do not think this should be a paper. I mean, I see it more as a technical report or, even better, as the preliminary part of a wider study, maybe from the selection of the best models to dedicated projections of hazard and impacts. I am aware that other studies dealing on the evaluations of newest GCM-RCM simulations do exist, focusing on small regions, e.g. the one cited by authors about Sardinia, but I feel that this is not a research paper, but a (very well performed) study on the performance of models on a test region. Thus, I am questioning myself: once the authors have decided that one combination of GCMs-RCMs performs better than the others, for each quantity analyzed (precipitation, temperature, drought), time scale (annual, seasonal), sub-region (3 for Sicily, 3 for Calabria).. what shall the reader do with this information inserted in a scientific paper? The region is very small, so - as the authors say (see lines 452-453) - the choice of the best model depends on many factors, making this piece of work not conclusive. What shall be really of interest is what the authors plan for further analyses (Lines 454-456). I also have another major point about the possible publication: the paper is not about droughts. Drought is just slightly touched and with very basic metrics, far from the current standard in drought-related analyses, so my final verdict is to reject this submission. However, I see that authors made great efforts, so they might consider to rethink about the paper and try to resubmit, but I would definitely remove the word droughts from title.

AC:     *We thank the referee for the attention devoted to our study and his partial appreciation of the manuscript. Indeed, as the referee mentioned, many papers deal only with the evaluation of RCM historical simulations and do not include assessment of future impacts of climate change, as confirmed by the bibliographic review in Table I. Furthermore, one of the distinguishing features of our study compared to the literature on the subject, is the high density of temperature and precipitation ground-based stations available in the case-study region; besides, the test region is representative of one of the main hot-spots for climate change – the Mediterranean Basin. Concerning its spatial extent (about 40,540 km²), it should be pointed out that our interest lies in the implementation of RCMs for climate change impact studies and hydrological applications at small spatial scale regions with a complex topography (see LL 47-50). To this end, it is particularly important to test the RCMs' skills in encompassing surface heterogeneities and mesoscale atmospheric processes at the considered spatial scale. We agree that the choice of the best model depends on many factors. That's exactly why our study intends to provide indications on the best model to choose based on the variable, the time and the spatial scale considered.*

*Moreover, it is worth highlighting the novelties introduced by the methodological approach, which adopts both PCA for identifying sub-regions in the analyzed area and proposes hybrid rankings involving precipitation, temperature, and drought characteristics.*

*Finally, a comprehensive evaluation of RCMs is an important resource for readers and potential users of the RCM data. There are several ways to use this information, and the authors will not surely cover all possible ways. So, we want to provide the readership with a tool they can use for their specific purposes. For our part, we notice that this study could be useful for hydrological applications, where the use of a limited but properly selected set of models can help to avoid unnecessary computational burden, or for other high-temporal resolution applications, where information about models' performance allow the user to narrow down the search domain for the most suitable projections.*

*Regarding the drought analysis, we agree with the reviewer that more analyses could have been carried out. Therefore, following his suggestions, we will extend the analysis to the seasonal data, following the*

*investigation on precipitation and temperature, and we will include an analysis on the return period of drought duration as well.*

**RC:** Why not using also Med-Cordex?

*AC: We could include Med-CORDEX data in our study. However, only a couple of models are currently available at the resolution used in this study (0.11°), thus we decided to focus on EURO-CORDEX only.*

**RC:** Are the Euro-CORDEX bias-adjusted? Why not using the bias-adjusted runs?

*AC: The EURO-CORDEX data in our study are not bias-adjusted. This is because the bias-adjustment is usually based on observed data (as a calibration procedure) and is particularly useful when RCMs are used for future projections. However, future projections are out of the scope of the present study, which addresses the evaluation of historical climate models simulations. The basic idea behind an evaluation study is to analyze the models' skill in simulating hydro-climatic processes against observations, rather than to correct the simulations with respect to them, as instead it could be required, for instance, in the case of climate impact studies.*

**RC:** I'd like to see more details on the station data, which could be potentially one of the most interesting parts of the study.

*AC: A Table with the most relevant information about the weather stations used in this study will be attached to the revised version of the manuscript.*

**RC:** Don't include equations in the core manuscript, move them all to supplementary materials.

*AC: As a matter of fact, there are only three equations in the original manuscript. In the revised manuscript, few equations will be added regarding the drought analysis. However, for the sake of readability, we prefer to keep them in the main text of the manuscript.*

**RC:** Drought part is very poor. Why not using, at least, the SPI and the SPEI? Also, the choice of quantities related to drought are not enough to justify the publication, I'd expect a lot more (frequency of events, intensity, severity, return periods, spatial aggregation, etc.) especially on monthly basis (not annual).

*AC: Thanks for this valuable suggestion. We agree with the referee that our work will benefit from more analyses on droughts, though we only partially agree with carrying out some of the analyses that he/she suggests. In particular, SPI and SPEI, by definition, follow a standard normal distribution. Hence long-term statistics (mean, standard deviation, etc.) are the same for the model and the observations. This feature hinders the possibility to use the considered error metrics and models' ranking to evaluate the models' performances, as in principle differences between the statistics derived from simulated and observed standardized drought index series could be primarily accounted for as sampling variability, rather than the*

*actual RCMs' skill in reproducing wet and dry conditions. It is for this reason, that we preferred to apply the theory of runs to precipitation data for drought identification.*

*To extend the drought analysis, as suggested by the referee, drought events will be also identified on seasonal precipitation values simulated for the period 1971-2000. Also, the return period of drought events of fixed duration computed on both annual and seasonal precipitation data will be included in the revised manuscript. All these changes will be addressed in the revised Methodology Section of the manuscript as follows:*

***Author's changes to the manuscript (LL 186-196):*** *"Drought events were identified on both annual and seasonal (DJF, MAM, JJA, SON) precipitation values simulated for the period 1971-2000, according to the theory of runs (Yevjevich, 1967). In particular, drought events were selected as the periods during which consecutive annual or seasonal values of precipitation did not exceed a given threshold, here assumed equal to the long term means of annual and seasonal data (a different threshold for each season). Once drought events were identified, the corresponding drought characteristics in each cell were determined. In particular, the following statistics for drought characteristics are considered hereafter to assess the models' performance:*

- *Maximum drought duration $L_{max}$: maximum length of periods with consecutive annual precipitation values below the threshold;*
- *Maximum drought accumulated deficit $D_{max}$: maximum of the sums of the differences between the threshold and the precipitation values along with the drought duration;*
- *Maximum drought intensity $I_{max}$: maximum of the ratio between drought accumulated deficit and duration;*
- *Return period of drought events of fixed duration.*

*Concerning the return period of drought events, let $E$ be a critical drought (e.g., a drought with duration $L$ equal to a fixed value). Assuming independence between consecutive drought events, the return period of drought event $E$ can be expressed as (Gonzales and Valdes, 2003; Cancelliere and Salas, 2004; Cancelliere and Salas, 2010; Bonaccorso et al., 2012):*

$$T_E = \frac{E[L] + E[L_n]}{P[E]} \tag{1}$$

*where $E[L]$ is the expected value of drought duration $L$ and $E[L_n]$ is the expected value of the non-drought duration $L_n$ and $P[E]$ is the probability of occurrence of a critical drought $E$, which can be determined once that the probability distribution function of the event $E$ is known.*

*Regarding the probability distribution of drought duration, let us consider a periodic stochastic hydrological variable denoted as $X_{v,\tau}$, where $v$ represents the year and $\tau$ represents the season (or the month). According to the theory of runs, drought duration $L$ is the number of consecutive time intervals (seasons) where $X_{v,\tau} \leq x_{0,\tau}$ is preceded and followed by at least one season where $X_{v,\tau} > x_{0,\tau}$, where $x_{0,\tau}$ is a threshold level representing water demand. The original variable can be replaced by a Bernoulli variable $Y_{v,\tau}$ such that:*

$$\begin{cases} Y_{v,\tau} = 0 \ if \ X_{v,t} \leq x_{0,\tau} \ (deficit) \\ Y_{v,\tau} = 1 \ if \ X_{v,t} > x_{0,\tau} \ (surplus) \end{cases} \tag{2}$$

*Assuming that $Y_{v,\tau}$ is a lag-1 Markov stationary process, it can be shown (Sen, 1976; Cancelliere and Salas 2004; Cancelliere and Salas, 2010) that the probability distribution of drought duration $L$ is geometric with parameter $p_{01}$:*

$$f_L(\ell) = P[L = \ell] = (1 - p_{01})^{\ell-1} p_{01} \tag{3}$$

The parameter $p_{01}$ represents the transition probability from a deficit to a surplus, namely $p_{01} = [Y_{v,\tau} = 1 | Y_{v,\tau-1} = 0]$.

Estimation of transition probabilities can be carried out following a non-parametric approach based on maximum likelihood, which leads to (Bonaccorso et al., 2012):

$$p_{01} = 1 - p_{00} = 1 - \frac{n_{00}}{n_{00} + n_{01}} \tag{4}$$

where $n_{00}$ is the number of observations $y_{v,\tau} = 0$, for which $y_{v,\tau-1} = 0$, and $n_{01}$ is the number of observations $y_{v,\tau} = 1$, for which $y_{v,\tau-1} = 0$.

For independent stationary series, the probability distribution of drought duration $L$ is geometric with parameter $p_1 = P[Y_\tau = 1]$. The latter can be simply estimated by applying a frequency analysis on $Y_\tau$.

Following previous studies (Bonaccorso et al., 2003; Cancelliere and Salas, 2004), the annual series were assumed independent stationary, whereas the seasonal series as lag-1 stationary Markov."

As an example of the application of the abovementioned methodology, the box-plots representing the frequency distribution of both observed and RCMs return periods for drought durations equal to 1, 3, 5 and 7 years for the annual series and drought durations equal to 2, 4, 6 and 8 seasons for the seasonal series are illustrated in Figures R2.1 and R2.2, respectively.

As expected, the absolute errors increase as the return period increases. As for the annual series (Figure R2.1), regardless of the return period, ECE-CCLM shows both the smallest IQR and median errors. Among the other models, ECE-RACM shows a median error close to 0 with a larger IQR than ECE-CCLM; on the contrary, Had-RACM IQR is similar to the one of ECE-CCLM but the model always overestimates the errors, with the only exception of Tr=1 year where the error is largely underestimated. It's worth pointing out that the range of errors in the plots at the top is usually much smaller than the considered return period. On the other hand, in the plots at the bottom, and particularly with Tr=7 years, the errors get too large, thus discouraging the use of RCMs for extremely long drought events analysis.

As for the seasonal series (Figure R2.2), the Had-CCLM and HAD-RACM provide the best performance in terms of smaller IQR and median errors for each considered return period. Other models showing limited errors are CM5-CCLM, Had-RCA4, the MPI models (except MPI-CCLM), and the Nor-HIRH. Once again, it's worth observing that the range of the errors increases significantly for Tr greater than or equal to 6 seasons (i.e. the plots at the bottom), leading to unreliable estimates.

Regarding the possibility to include spatial aggregation of drought events (or drought characteristics) in our study, as the current investigations rest upon at-site analysis, for the sake of clarity (and brevity) we prefer to not introduce results at different spatial levels. However, we aim to consider regional droughts in a future evaluation study.

**RC:** Some conclusions are exactly what one might expect: precipitation is modelled worse than temperature, drought (as computed in this study) is similar to precipitation, RCMs deeply affect the results more than GCMs.

**AC:** *In light of the changes made to the manuscript, and in particular, of the enhanced drought analysis, conclusions will be rewritten, highlighting the main novelties of the study.*

[Figure]

*Figure R2.1. Box-plots representing the frequency distribution of RCMs errors in return period for the annual series and the whole study area*

[Figure]

*Figure R2.2. Box-plots representing the frequency distribution of RCMs errors in return period for the seasonal series and the whole study area*

*References to be added in the new version of the manuscript:*

*Bonaccorso, B., Cancelliere, A., and Rossi, G. (2003). An analytical formulation of return period of drought severity. Stochast. Environ. Res. Risk Assess., 17, 157–174.*

*Bonaccorso, B., Cancelliere, A. and Rossi, G. (2012). Methods for Drought Analysis and Forecasting. In: Methods and Applications of Statistics in the Atmospheric and Earth Sciences. p. 150-184, Hoboken, John Wiley and Sons, ISBN: 9780470503447.*

*Cancelliere, A. and Salas, J. (2004). Drought length properties for periodic stochastic hydrological data. Water Resourc. Res., 10, 1–13.*

*Cancelliere, A. and Salas, J. (2010). Drought probabilities and return period for annual streamflows series. J. Hydrol., 391, 77–89.*

*Gonzalez, J. and Valdes, J. (2003). Bivariate drought recurrence analysis using tree ring reconstructions. J. Hydrol. Eng., 8, 247–258.*

*Sen, Z. (1976). Wet and dry periods of annual flow series. J. Hydraul. Div., 102, 1503–1514.*

*Yevjevich, V. (1967). An Objective Approach to Definitions and Investigations of Continental Hydrologic Droughts. Hydrology Paper 23, Colorado State University, Fort Collins, CO.*

---

## Author Response (AR1)

*We warmly thank the Editor and the Referees for their careful reading of the manuscript and their valuable comments. Please find below our answers (in italic) to all the items raised, and the tracked-changes version, to whom the line numbers refer.*

*Please note that in the revised version of the paper, numbering and positioning of the figures were reorganized, joining some graphs that were previously presented separately, with the aim to include more results (especially about drought characteristics, as required by Anonymous Referee #2) and increase remarkably the information content of the paper, keeping its overall structure as clear as possible.*

**Anonymous Referee #1**

**RC:** This paper quantifies the performances of several combinations of regional climate models (RCMs) driven by different general circulation models (GCMs) in two regions of southern Italy. The GCM-RCM combinations are part of the Coordinated Regional Climate Downscaling Experiment (CORDEX) initiative in the European domain (EURO- CORDEX). Performances are evaluated in the ability to capture the spatial variability of mean annual and seasonal precipitation (P) and temperature (T), as well as three drought metrics derived by applying the run theory. I enjoyed reading this paper, which is well written; presents rigorous analyses; critically discusses the results; and has practical utility for impact studies in the study regions, since it provides a list of best GCM-RCM combinations. I recommend its publication and I only have a few minor requests and suggestions.

*AC: We thank the referee for the positive comments.*

**RC:** The authors should provide more details on how they applied the principal component analysis (PCA): Was it applied on monthly or annual P? The PCA returns spatial patterns that explain most of the P variability. How did the authors derive the subregions within these spatial patterns? This should be clarified.

*AC: We thank the referee for this comment, which allowed us to improve the description of the PCA. At this regard, further information to the manuscript was added as follows:*

[revised manuscript text omitted]

**RC:**   When ranking the models based on performance in reproducing annual P, the authors find that nine models have similar error metrics. Have they tried to compute the mean rank of each model across the zones and even across the five time scales considered (annual and the four seasons)? There may be some models that are consistently in the top (lowest ranks) and these should be mentioned.

*AC:    This aspect was already partially investigated in a previous version of the manuscript, where we pointed out that the models Had-RACM, ECE-CCLM and Had-CCLM have the overall better performances at both annual and seasonal time scales. Thus, we have enlarged and made clearer the discussion in the revised version of the manuscript, adding also a figure (new Fig. 15) highlighting deviations in the performances of some models (e.g., CM5-ALAD), considering both the annual scale and the average behaviour at the seasonal scale (the higher the deviation, the higher the distance from the bisector).*

*Author's changes to the manuscript (LL 559-565):* *"Figure 15 shows a comparison between the ranking of interannual variability of annual precipitation and the average position in the ranking of seasonal precipitation. It highlights possible deviations of the performances of the models at different time scales (the higher the deviation, the higher the distance from the bisector). When considering the seasonal scale, the reduced performance of CM5-ALAD is evident, such as the better ranking of MPI-CCLM. In general, the best models both at the interannual and the seasonal scale, are Had-RACM and ECE-CCLM, followed by the two versions of ECE-RACM and two other CCLM models (namely, MPI-CCLM and Had-CCLM, the latter being penalized by the relatively lower ranking in winter)."*

[Figure]

*Figure 15. Comparison between the RCM position in the ranking of interannual variability of annual precipitation versus the average position in the ranking of seasonal variability of seasonal precipitation. Data concerns the whole study area (Calabria and Sicily).*

**RC:** Line 45: Extremes occur everywhere. I suggest changing to "… occurrence of particularly intense extreme events, …". If this is what the authors mean, a reference is also needed.

*AC:* *The sentence was changed according to the suggestion.*

*Author's changes to the manuscript (LL 48-50):* *"… and is characterized by the occurrence of particularly intense extreme eventss, such as prolonged droughts and high-intensity storms leading to floods (Bonaccorso et al., 2013; Bonaccorso et al., 2015a and 2015b; Llasat et al., 2016; Senatore et al., 2020)."*

**RC:** Line 55: CMIP5 has been already defined; just use the acronym.

**RC:** Line 61: I suggest adding "historical" before simulations.

**RC:** Line 326: it should be "show".

*AC:* *The text was revised accordingly.*

**Anonymous Referee #2**

**RC:** The paper is well written, the state-of-art well described in the introduction, and the methodology used in this study are clear and can be easily understood from the paper. Overall, the quality of this paper is good, but to be honest I do not think this should be a paper. I mean, I see it more as a technical report or, even better, as the preliminary part of a wider study, maybe from the selection of the best models to dedicated projections of hazard and impacts. I am aware that other studies dealing on the evaluations of newest GCM-RCM simulations do exist, focusing on small regions, e.g. the one cited by authors about Sardinia, but I feel that this is not a research paper, but a (very well performed) study on the performance of models on a test region. Thus, I am questioning myself: once the authors have decided that one combination of GCMs-RCMs performs better than the others, for each quantity analyzed (precipitation, temperature, drought), time scale (annual, seasonal), sub-region (3 for Sicily, 3 for Calabria).. what shall the reader do with this information inserted in a scientific paper? The region is very small, so - as the authors say (see lines 452-453) - the choice of the best model depends on many factors, making this piece of work not conclusive. What shall be really of interest is what the authors plan for further analyses (Lines 454-456). I also have another major point about the possible publication: the paper is not about droughts. Drought is just slightly touched and with very basic metrics, far from the current standard in drought-related analyses, so my final verdict is to reject this submission. However, I see that authors made great efforts, so they might consider to rethink about the paper and try to resubmit, but I would definitely remove the word droughts from title.

**AC:** *We thank the referee for the attention devoted to our study and his partial appreciation of the manuscript. As the referee mentioned, many papers deal only with the evaluation of RCM historical simulations and do not include assessment of future impacts of climate change, as confirmed by the bibliographic review in Table I. Nonetheless, one of the distinguishing features of our study compared to the literature on the subject, is the high density of temperature and precipitation ground-based stations available in the case-study region. Besides, the target region is representative of one of the main hot-spots for climate change – the Mediterranean Basin. Concerning its spatial extent (about 40,540 km²), it should be pointed out that our interest lies in the implementation of RCMs for climate change impact studies and hydrological applications at small spatial scale regions with a complex topography (see LL 51-54). To this end, it is particularly important to test the RCMs' skills in encompassing surface heterogeneities and mesoscale atmospheric processes at the considered spatial scale. We agree that the choice of the best model depends on many factors. That's exactly why our study intends to provide indications on the best model to choose based on the variable, the time and the spatial scale considered.*

*Moreover, it is worth highlighting the novelties introduced by the methodological approach, which adopts both PCA for identifying sub-regions in the analyzed area and proposes hybrid rankings involving precipitation, temperature, and drought characteristics.*

*Finally, a comprehensive evaluation of RCMs is an important resource for readers and potential users of the RCM data. There are several ways to use this information, and the authors will not surely cover all possible ways. So, we want to provide the readership with a tool that they can use for specific purposes. From our part, we notice that this study could be useful for hydrological applications, where the use of a limited but properly selected set of models can help to avoid unnecessary computational burden, or for other high-temporal resolution applications, where information about models' performance allow the user to narrow down the search domain for the most suitable projections.*

*Regarding the drought analysis, we agree with the reviewer that more analyses were needed. Therefore, following his/her suggestions, we extended the analysis to the seasonal data, likewise the investigation on*

*precipitation and temperature, and to the return period of drought duration as well (new Section 4.3 and related discussion in improved Sections 5.1 and 5.2). The effects of the improved drought analysis also influence the overall ranking (Section 5.4)*

**RC:** Why not using also Med-Cordex?

*AC: We could include Med-CORDEX data in our study. However, only a couple of models are currently available at the resolution used in this study (0.11°), thus we decided to focus on EURO-CORDEX only.*

**RC:** Are the Euro-CORDEX bias-adjusted? Why not using the bias-adjusted runs?

*AC: The EURO-CORDEX data in our study are not bias-adjusted. This is because the bias-adjustment is usually based on observed data (as a calibration procedure) and is particularly useful when RCMs are used for future projections. However, future projections are out of the scope of the present study, which addresses the evaluation of historical climate models simulations. The basic idea behind an evaluation study is to analyze the models' skill in simulating hydro-climatic processes against observations, rather than to correct the simulations as it is, for instance, required in the case of climate impact studies.*

**RC:** I'd like to see more details on the station data, which could be potentially one of the most interesting parts of the study.

*AC: A Table with the most relevant information about the weather stations used in this study has been added as Supplementary Material.*

**RC:** Don't include equations in the core manuscript, move them all to supplementary materials.

*AC: As a matter of fact, there are only three equations in the original manuscript. In the revised manuscript, few equations were added regarding the drought analysis, increasing the total number to eight. For the sake of readability, we prefer to keep them in the main text of the manuscript.*

**RC:** Drought part is very poor. Why not using, at least, the SPI and the SPEI? Also, the choice of quantities related to drought are not enough to justify the publication, I'd expect a lot more (frequency of events, intensity, severity, return periods, spatial aggregation, etc.) especially on monthly basis (not annual).

*AC: We thank the referee for this valuable suggestion. We agree with the referee that our work can benefit from more analyses on droughts, though we only partially agree with carrying out some of the analyses that he/she suggests. In particular, SPI and SPEI, by definition, follow a standard normal distribution. Hence long-term statistics (mean, standard deviation, etc.) are the same for the model and the observations. This feature hinders the possibility to use the considered error metrics and models' ranking to evaluate the models' performances, as in principle differences between the statistics derived from simulated and observed standardized drought index series could be primarily accounted for as sampling variability, rather than the*

*actual RCMs' skill in reproducing wet and dry conditions. For this reason we preferred to apply the theory of runs to precipitation data for drought identification.*

*To extend the drought analysis, as suggested by the referee, drought events were also identified on seasonal precipitation values simulated for the period 1971-2000. Also, the return period of drought events of fixed duration computed on both annual and seasonal precipitation data was included in the revised manuscript. The methodological aspects of these changes were addressed in the revised Section 3.2 of the manuscript (LL 239-329). Concerning results, the following outcomes are described and discussed:*

- *Frequency distribution of RCMs percentage errors in maximum drought duration, maximum drought accumulated deficit and maximum drought intensity at the annual and seasonal scale (new Fig. 11);*
- *Frequency distribution of RCMs percentage errors in the return period of drought event of duration L equal to 1, 3, 5 and 7 years (new Fig. 12);*
- *Frequency distribution of RCMs percentage errors in the return period of drought event of duration L equal to 2, 4, 6 and 8 seasons (new Fig. 13);*
- *Improved rankings based on drought characteristics merging drought maximum intensities and return periods at the annual and seasonal levels (new Fig. 14);*
- *Improved overall ranking embedding the new drought analysis (new Fig. 16).*

*All the new findings were introduced and described in the improved sections concerning drought characteristics results (Section 4.3.1 LL 470-489 and the whole new Section 4.3.2 about results at the seasonal scale) and discussion (particularly Section 5.1, LL 566-576, and Section 5.2, LL 601-611). Tables 5 and 6 were also updated accordingly.*

*Finally, regarding the possibility to include spatial aggregation of drought events (or drought characteristics) in our study, as the current investigations rest upon at-site analysis, for the sake of clarity (and brevity) we preferred to not introduce results at different spatial levels. However, we aim to consider regional droughts in a future evaluation study.*

**RC:** Some conclusions are exactly what one might expect: precipitation is modelled worse than temperature, drought (as computed in this study) is similar to precipitation, RCMs deeply affect the results more than GCMs.

**AC:** *In light of the changes made to the manuscript, and in particular, of the enhanced drought analysis, conclusions were rewritten, highlighting the main findings of the study.*

[revised manuscript text omitted]

---

## Author Response (AR2)

*Once more, we warmly thank the Editor and the Referee #3 for careful reading the manuscript and their valuable comments. Please find below our answers (in italic) to all the items raised, and the tracked-changes version, to whom the line numbers refer.*

**Editor**

Dear Authors,

We have finally received a sufficient number of reviews. As you can see, there are still some minor concerns related to the matching between the content of the paper and the title.

I overall agree with the reviewer on that, especially given the expected inclusion of the paper as part of a special issue on drought. However, I'm confident that the authors can make a last additional effort to better highlight the relevance of the study for the community of drought researches.

I'm looking forward to receiving your revised manuscript for a final decision on its publications.

Best regards,

Carmelo Cammalleri

*AC:      As replied to Referee #3, we changed the title of the manuscript in: "Evaluation of EURO-CORDEX historical simulations by high-quality observational datasets in Southern Italy: insights on drought assessment". Furthermore, we have added further short sentences that attempt to frame better the relevance of our study for drought research (LL 17, 35, 88).*

**Anonymous Referee #3**

The manuscript titled "Towards a reliable assessment of climate change impact on droughts in Southern Italy: Evaluation of EURO-CORDEX historical simulations by high-quality observational datasets" by Peres et al.

The paper fits within the stated scope of the journal. The authors carefully addressed most of the comments from two reviewers and made appropriate changes in the manuscript. The present version of the manuscript has been improved following reviewers' suggestions.

*AC:     We thank the referee for the positive comments.*

There are still two minor comments:

1. The title is not very appropriate. I appreciate that the authors have added more analysis for droughts. But to be honest, there still quite a large part of the paper is about precipitation and temperature, not only drought. So either the authors need to cut some results and add/focus more on the droughts (at least this should be the majority part of the paper), or they need to change the title.

*AC:     We changed the title of the manuscript in: "Evaluation of EURO-CORDEX historical simulations by high-quality observational datasets in Southern Italy: insights on drought assessment". Furthermore, we added further short sentences (LL 17, 35, 88) backing up the topic of drought. Indeed, precipitation and temperature performance analysis are necessary preliminary steps to drought performance assessment.*

2. The paper has done a tremendous analysis for different zones, but there are no information or results about the zones in the conclusion?

[revised manuscript text omitted]